# Unveiling the Spectrum of Minor Genes in Cardiomyopathies: A Narrative Review

**DOI:** 10.3390/ijms25189787

**Published:** 2024-09-10

**Authors:** Caterina Micolonghi, Federica Perrone, Marco Fabiani, Silvia Caroselli, Camilla Savio, Antonio Pizzuti, Aldo Germani, Vincenzo Visco, Simona Petrucci, Speranza Rubattu, Maria Piane

**Affiliations:** 1Department of Experimental Medicine, Faculty of Medicine and Dentistry, Sapienza University of Rome, 00161 Rome, Italy; caterina.micolonghi@uniroma1.it (C.M.); fede.perrone@gmail.com (F.P.); marco.fab88@gmail.com (M.F.); caroselli.silvia@gmail.com (S.C.); antonio.pizzuti@uniroma1.it (A.P.); 2Department of Neuroscience, Istituto Superiore di Sanità, 00161 Rome, Italy; 3ALTAMEDICA, Human Genetics, 00198 Rome, Italy; 4Juno Genetics, Reproductive Genetics, 00188 Rome, Italy; 5S. Andrea University Hospital, 00189 Rome, Italy; camilla.savio@gmail.com (C.S.); vincenzo.visco1@uniroma1.it (V.V.); simona.petrucci@uniroma1.it (S.P.); maria.piane@uniroma1.it (M.P.); 6Medical Genetics Unit, IRCCS Mendel Casa Sollievo della Sofferenza, 71013 San Giovanni Rotondo, Italy; 7Department of Clinical and Molecular Medicine, Faculty of Medicine and Psychology, Sapienza University of Rome, 00189 Rome, Italy; aldo.germani@uniroma1.it; 8IRCCS Neuromed, 86077 Pozzilli, Italy

**Keywords:** cardiomyopathies, genetics, ACM, ARVC, DCM, HCM

## Abstract

Hereditary cardiomyopathies (CMPs), including arrhythmogenic cardiomyopathy (ACM), dilated cardiomyopathy (DCM), and hypertrophic cardiomyopathy (HCM), represent a group of heart disorders that significantly contribute to cardiovascular morbidity and mortality and are often driven by genetic factors. Recent advances in next-generation sequencing (NGS) technology have enabled the identification of rare variants in both well-established and minor genes associated with CMPs. Nowadays, a set of core genes is included in diagnostic panels for ACM, DCM, and HCM. On the other hand, despite their lesser-known status, variants in the minor genes may contribute to disease mechanisms and influence prognosis. This review evaluates the current evidence supporting the involvement of the minor genes in CMPs, considering their potential pathogenicity and clinical significance. A comprehensive analysis of databases, such as ClinGen, ClinVar, and *GeneReviews*, along with recent literature and diagnostic guidelines provides a thorough overview of the genetic landscape of minor genes in CMPs and offers guidance in clinical practice, evaluating each case individually based on the clinical referral, and insights for future research. Given the increasing knowledge on these less understood genetic factors, future studies are essential to clearly assess their roles, ultimately leading to improved diagnostic precision and therapeutic strategies in hereditary CMPs.

## 1. Introduction

Cardiomyopathies (CMPs) represent a heterogeneous group of heart disorders characterized by structural and functional abnormalities of the myocardium, often stemming from genetic factors. Among these, arrhythmogenic cardiomyopathy (ACM, MIM 610476, 610193, 607450, 609040, 604400), dilated cardiomyopathy (DCM, MIM 613426, 601494, 613881, 604765, 115200, 613172, 601154, 611879, 604145), and hypertrophic cardiomyopathy (HCM, MIM 612098, 115197, 192600, 608758, 608751, 613690, 115195, 115196) stand out as significant contributors to cardiovascular morbidity and mortality. Understanding the intricate genetic underpinnings of these conditions is paramount for advancing diagnostic precision, prognostic assessment, and targeted therapeutic interventions. Over the years, substantial progress has been made in unraveling the complex genetic landscape associated with cardiomyopathies [1]. Indeed, recent advances in next-generation sequencing (NGS) technology have enabled the identification of rare genetic variants, both in genes clearly associated with the disease and in minor genes that show a lower correlation to the clinical phenotype [2,3]. Panel genes, whole-exome sequencing (WES), and whole-genome sequencing (WGS) represent significant advancements in the genetic evaluation of CMPs. Panel genes and WES, which focus on protein-coding regions, effectively identify pathogenic variants and incidental findings. In contrast, WGS provides comprehensive genome coverage, detecting both intronic and structural variations. Despite their relevance for early diagnosis and personalized treatment, variables linked to data interpretation, high costs, and clinical parameters still limit their full application [3].

Nowadays, thanks to NGS technology, which has enabled the identification of pathogenic variants in various genes, and functional and genetic evidence that has clarified their clinical significance, several genes have been confirmed as definitively associated with ACM, HCM, and DCM [4,5,6]. ACM is a cardiac disorder characterized by the progressive replacement of myocardial tissue with fibrous and fatty deposits [7]. Currently, ACM includes several phenotypic variants: (i) the classical right ventricular dominant form (arrhythmogenic right ventricular cardiomyopathy, ARVC), which particularly affects the right ventricle, (ii) the left ventricular-dominant form (arrhythmogenic left ventricular cardiomyopathy, ALVC), which impairs the left ventricle with no or minor right ventricle abnormalities, and (iii) biventricular cardiomyopathy (BiVACM), which is characterized by the parallel involvement of both ventricles [8]. The pathogenesis of ACM involves a complex interplay of genetic and environmental factors. Pathogenic variants affecting desmosomal proteins, crucial for cell-to-cell adhesion in the myocardium, are commonly implicated in ACM [9,10]. According to the sources cited above, five genes are classified as having definitive evidence of associations with ACM: *DSC2* [11] (MIM 125645)*, DSG2* [12] (MIM 125671)*, DSP* [13] (MIM 125647)*, PKP2* [14] (MIM 602861), and *TMEM43* [15] (MIM 612048). DCM is a cardiac disorder characterized by the enlargement and weakened pumping ability of the left ventricle, leading to decreased contractility and impaired systolic function [16]. Genetic factors play a significant role in DCM, with pathogenic variants in different genes associated with the regulation of myocardial structure and function [17]. While there is not a fixed set of “core genes” for DCM, several genes have been associated, according to the aforementioned sources, as having definitive evidence of associations with *DCM: BAG3* [18] (MIM 603883)*, DES* [19] (MIM 125660)*, FLNC* [20] (MIM 102565)*, LMNA* [21] (MIM 150330)*, RBM20* [22] (MIM 613171)*, SCN5A* [23] (MIM 600163)*, TNNC1* [24] (MIM 191040) and *TTN* [25] (MIM 188840). HCM is a primary, often inherited, cardiac disease characterized by abnormal hypertrophy of the heart, primarily affecting the left ventricle. The fundamental pathogenesis of HCM involves a genetic mutation leading to the abnormal hypertrophy of cardiomyocytes, resulting in increased cardiac mass [26]. HCM is primarily linked to pathogenic variants in genes encoding sarcomere proteins, which play a crucial role in cardiac muscle contraction. Each gene serves as a critical element in the sarcomere, influencing muscle contraction, structural integrity, and the overall physiological balance within the heart. Nowadays, according to the aforementioned sources, eight core genes are classified as having definitive evidence of associations with HCM: *ACTC1* (MIM 102540)*, MYBPC3* [27] (MIM 600958)*, MYH7* [28] (MIM 160760)*, MYL2* [29] (MIM 160781)*, MYL3* [30] (MIM 160790, *TNNI3* [31] (MIM 191044)*, TNNT2* [32] (MIM 191045), *TPM1* [33] (MIM 191010), and *FHOD3* [34] (MIM 609691) [35].

Based on the literature, minor genes in CMPs can be considered genes of uncertain significance (GUSs) due to limited evidence of pathogenicity, difficulties in interpreting variants, phenotype variability, and lack of robust functional and segregation studies. Therefore, GUSs raise several challenges in cardiogenetics, as they often harbor variants whose effects on protein function and disease risk are not yet well understood. In fact, even though current research has begun to explore the potential involvement of these minor genes (e.g., *TRIM63* MIM 606131) in CMPs [36,37], their potential contribution to disease mechanisms is still unclear. Functional studies and segregation analysis are critical to determine their potential pathogenic role and clinical significance in CMPs.

The genetic tests for CMPs often consist of extensive gene panels that include both genes closely linked to various clinical phenotypes, such as HCM, DCM, and ACM, as well as genes with lower associations and genotype–phenotype correlations that are not yet definitively established. While broader panels increase the likelihood of identifying new genes associated with the different forms of genetically based CMPs, they have the disadvantage of detecting variants of uncertain significance (VUS) much more frequently than more specific panels (87% vs. 30%), complicating the interpretation of genetic testing and the clinical management of CMPs [38]. This review aims: (i) to provide a comprehensive synthesis of current knowledge on the less understood, non-definitive genes devoid of a clear association with syndromic disorders, involved in CMPs—our effort fulfills the need to unveil the potential role of the rare genetic contributors to ACM, DCM, and HCM; and (ii) to provide guidance on the preparation of genetic test reports in clinical practice, evaluating on a case-by-case basis, according to the clinical referral, the genes and the gene variants to report to clinicians, excluding GUSs, for the proper management of both the proband and family members.

### 1.1. Identification of Reported Minor CMP Genes

The genes not yet definitively associated with CMPs were selected through a systematic review of the literature and querying different sources in the context of the three CMP phenotypes (ACM, DCM, and HCM). We focused on genes considered to have moderate or limited association with CMPs according to the investigated sources. Several caveats deserve to be made. Regarding ACM, this term has recently been proposed to include the broader spectrum of phenotypic variants [39] and is therefore used as the most up-to-date designation here. However, disease–gene curation is available only under the original designation of “ARVC” in ClinGen [40,41], *GeneReviews* [42], and the European Molecular Genetics Quality Network (EMQN) [43,44] by embracing the description of the other forms (MONDO:0016587 and ORPHA:247). As of today, there is a lack of concordance for *JUP* (MIM 173325) and *PLN* (MIM 172405). While they are considered definitive genes for ACM and HCM/DCM, respectively, in EMQN [43,44], they are not reported as definitive in ClinGen [45] because their “gene–disease validity” has not been fully assessed. However, the literature is overwhelmingly supportive of their role in CMPs [46]. Since this review focuses on minor genes, *JUP* and *PLN* will not be discussed. The same holds true for *ALPK3* (MIM 617608). Although *ALPK3* is definitive for HCM in ClinGen and *GeneReviews*, this gene is not considered in the other sources. Therefore, after excluding genes with clearly definitive associations with CMPs and genes associated with syndromic conditions, 41 genes will be discussed as minor genes in this article.

### 1.2. Gathering Sources

To gain a comprehensive view of the minor genes associated with the three forms of CMPs discussed herein, we used ClinGen as a primary source [45]. It provides an authoritative and centralized resource that defines the clinical relevance of the genes and variants to be used in precision medicine and research. ClinGen uses various methodologies to define the association between rare variants and Mendelian diseases such as CMPs. These methods include segregation of variants with disease in large family pedigrees, case–control studies, and functional studies to analyze the consequences of variants. ClinGen’s approach to determining “gene–disease validity” involves a meticulous review process to assess the strength of evidence linking rare variants to CMPs. These classifications are based on the careful analysis of published genetic and experimental data by data curators and expert panels. They generate strong, moderate, or limited levels of evidence that can be further modified to definitive or even refuted once new evidence becomes available.

The second source used was ClinVar [47]. ClinVar is a public repository maintained by the National Center for Biotechnology Information (NCBI) [48], part of the National Library of Medicine (NLM) [49] at the National Institutes of Health (NIH) [49,50]. ClinVar accepts and archives genetic variants identified in different clinical phenotypes by diagnostic laboratories, researchers, clinicians, and organizations, allowing for the re-evaluation of classifications over time. More specifically, ClinVar supports standard terms for variant classifications recommended by the ACMG/AMP [51]. This standard includes five levels—benign (B), likely benign (LB), uncertain significance (VUS), likely pathogenic (LP), and pathogenic (P)—to describe the pathogenicity of germline variants for Mendelian disorders based on criteria using several types of evidence (e.g., population data, computational data, functional data, segregation data). This should be also used by authors, submitting their data to ClinVar and PubMed.

*GeneReviews* [42], a journal edited by the University of Washington, Seattle, was also used as a resource. It aims to provide clinically relevant and actionable information for hereditary conditions to associate genes with specific diseases by collecting information from different sources, including peer-reviewed and research studies, data from clinical experts and researchers, genetic databases such as OMIM (Online Mendelian Inheritance in Man) [52], and other genomic databases.

To complete the evaluation of minor genes, we also considered the positions of the EMQN [43,44], as outlined in their 2023 recommendations for genetic testing in hereditary CMPs and arrhythmias [43]. Additionally, the 2022 Expert Consensus Statement on the State of Genetic Testing for Heart Disease provides valuable insights and guidance [53].

### 1.3. Literature Review

A comprehensive review of articles was conducted using PubMed as the primary database, searching for the publication period from 1990 to 2024. The keywords used were: “arrhythmogenic cardiomyopathy,” “dilated cardiomyopathy,” “hypertrophic cardiomyopathy,” and specific gene names. These terms were used in various combinations to maximize the retrieval of relevant articles. A filter was applied to narrow down the search results to articles written in English only to ensure that the articles were based on widely accessible sources and accepted by the international research community. The selection process involved reviewing the titles and abstracts of the retrieved articles to assess their relevance to the topic. Priority was given to articles that provided novel insights into the genetic underpinnings of ACM, DCM, and HCM based on functional studies, segregation in families, and clinical implications of genetic findings.

## 2. Gene Classification

In this review the reported genes are classified as definitive, moderate, limited, not associated, not curated, according to ClinGen terminology (Figure 1). The associations with different forms of CMPs (ACM, DCM, and HCM) are reported in Table 1 for definitive genes and in Table 2 for minor genes based on the classification provided by the cited sources. Genes considered “strong” in *GeneReviews* and the Expert Consensus were classified as definitive, and those considered “moderate intrinsic” in the Expert Consensus were classified as moderate, according to the ClinGen classification. Moreover, chromosomal location and mode of inheritance for each gene are provided in the corresponding tables. Most CMPs are inherited in an autosomal dominant (AD) manner (i.e., caused by a monoallelic variant in a single gene), often with incomplete penetrance and variable expressivity. Autosomal recessive (AR; i.e., caused by biallelic variants in a single gene) and autosomal semidominant (SD; may manifest in both monoallelic and biallelic states) forms are also described.

### 2.1. Definitive Genes Associated with ACM, DCM, and HCM

The following table lists the definitive genes that have been identified in association with ACM (ARVC, ALVC, and/or BiVACM), DCM, and HCM according to: ClinGen [45], GeneReviews [42], EMQN 2023 [43,44], and a 2022 expert consensus statement [53]. 

### 2.2. Minor Genes in ACM, DCM, and HCM

#### 2.2.1. ABCC9

The ATP-binding cassette subfamily C member 9 gene (*ABCC9*, MIM 601439), which encodes the cardiac sulfonylurea receptor subunit SUR2A of the cardiac KATP channel, plays a critical role in linking cellular metabolism to electrical activity through the regulation of potassium flow. These channels are vital in adapting the membrane potential to reflect the cellular ATP/ADP ratio, influencing cardiac excitability and protection under metabolic stress [54,55]. Mutations in *ABCC9* are directly associated with DCM. Initial discoveries highlighted two specific mutations in exon 38 based on a genetic screening of 323 DCM patients. One is a frameshift mutation (c.4570_4572delinsAAAT, p.(Leu1524fs)) and the second one is a missense mutation (c.4537G>A, p.(Ala1513Thr)), both altering KATP channel function in vitro and suggesting a mechanistic link to DCM pathogenesis [56]. Further research identified additional variants: the c.4517_4527del, p.(Arg1506fs) in Mexican DCM patients, and p.(Lys976Ile) and p.(Arg1197Cys) in a Chinese cohort, underscoring the global relevance of these mutations in DCM pathology [56,57,58]. In-depth studies demonstrated that these mutations disrupt the regulatory mechanisms of the KATP channel, leading to aberrant channel gating and impaired response to metabolic changes, which are crucial during cardiac stress [59]. Moreover, a recent case highlighted a loss-of-function (LOF) variant in *ABCC9* associated with severe ventricular arrhythmias and DCM, further validating the significant role of this gene in cardiac electrophysiology and its impact on cardiac disease phenotypes [60]. This evidence suggests that both gain-of-function (GOF) and LOF mutations in *ABCC9* can lead to diverse cardiac abnormalities, from Cantu syndrome to Brugada syndrome, emphasizing the complex role of KATP channels in heart diseases.

#### 2.2.2. ACTN2

The alpha-actinin 2 gene (*ACTN2*, MIM 102573) encodes a crucial sarcomere protein involved in linking and cross-linking actin filaments within the cardiac Z-disc, playing a significant role in the structural and functional integrity of the heart muscle. *ACTN2* mutations were associated with various forms of CMPs, including DCM and HCM, underscoring its vital role in cardiac pathophysiology [61,62]. Genomic studies identified several *ACTN2* mutations linked to CMPs. Notably, a missense mutation p.(Ala119Thr) was found to segregate with HCM in an Australian family, and additional missense variants p.(Thr495Met), p.(Glu583Ala), and p.(Glu628Gly) were identified in other HCM families, suggesting a genetic predisposition to HCM [63]. In the context of DCM, a specific *ACTN2* variant, p.(Gln9Arg), was associated with early mortality, emphasizing this gene’s impact on cardiac function [64]. Another study identified a missense variant p.(Leu320Arg) in all affected members of a Chinese family with DCM, further implicating *ACTN2* in the familial form of the disease [63,65]. Functional studies using hiPSC-derived cardiomyocytes from an HCM patient carrying a missense variant p.(Thr247Met) demonstrated cellular hypertrophy and sarcomere disorganization. These findings provide compelling evidence in favor of a role of *ACTN2* in HCM etiology [62]. Additionally, recent structural analyses showed that *ACTN2* mutations like p.(Ala119Thr), p.(Met228Thr), and p.(Thr247Met) disrupt actin-binding interactions, which are crucial for maintaining Z-disc structural integrity and by extension normal cardiac muscle function [66]. Moreover, *ACTN2* mutations can also play a role in the development of idiopathic RCM [67] and have been recently described in patients with ACM [68]. In addition, variants in *ACTN2* not only lead to structural CMPs but also have implications for cardiac arrhythmias, likely through mechanisms involving disrupted ion channel interactions within cardiomyocytes [69]. In summary, *ACTN2* mutations are significantly associated with both DCM and HCM with a complex phenotype spectrum.

#### 2.2.3. ANKRD1

The ankyrin repeat domain 1 gene (*ANKRD1,* MIM 609599) encodes the cardiac ankyrin repeat protein (CARP), which is involved in maintaining sarcomere integrity, myofibrillar signaling, and stretch sensing in the heart. This highly conserved protein interacts with other sarcomere proteins such as titin/connectin and myopalladin, playing a pivotal role in cardiomyocyte function [70]. Studies have demonstrated the involvement of *ANKRD1* in both HCM and DCM, with distinct genetic underpinnings that suggest a complex mechanism of action. In HCM, specific *ANKRD1* mutations, such as p.(Pro52Ala), p.(Thr123Met), and p.(Ile280Val), increase the binding affinity of CARP for titin/connectin and myopalladin. This alteration may disrupt normal myofibrillogenesis and sarcomere function, contributing to the hypertrophic response seen in HCM patients [71]. A study identified significant repressor activity alterations and hypertrophic responses in engineered heart tissue carrying the abovementioned HCM-associated *ANKRD1* mutations, supporting the pathogenic involvement of these variants [72]. *ANKRD1* appears to be more involved in DCM. *ANKRD1* coding regions were sequenced in DCM patients. As a result, a total of eight variants in nine patients were identified (three of them with family history) [73,74]. Expression studies performed in myoblastoid cell lines found loss of CARP binding with talin 1 and altered mechanical stretch-based signaling [74]. Mutant CARP proteins transfected in rat neonatal cardiomyocytes led to both significantly less repressor activity and greater phenylephrine-induced hypertrophy, suggesting altered function of CARP mutant protein [73].

#### 2.2.4. CALR3

The calreticulin 3 gene (*CALR3*, MIM 611414), encoding a member of the calcium-binding chaperones localized in the endoplasmic reticulum, has a controversial role in HCM. It is primarily expressed in the reproductive system, and its involvement in CMPs is based on limited and inconclusive evidence. Two heterozygous missense variants, p.(Arg73Gln) and p.(Lys82Arg), were identified in a study involving 252 unrelated HCM patients. However, the presence of multiple potentially disease-causing variants in other genes within the same patient cohort did not allow the interpretation of the *CALR3* variants in this context [75]. Subsequent studies have sought to clarify the role of *CALR3*, with one notable study assessing the gene influence in a large Dutch cohort of CMP patients. This study identified 17 rare heterozygous *CALR3* variants in 48 probands, and initially supported a higher prevalence of *CALR3* variants in CMP patients compared to controls. However, after accounting for a potential Dutch founder variant, the statistical significance disappeared, indicating that *CALR3* variants might not uniquely influence CMP risk [76]. Moreover, functional studies did not support a pathogenic role for *CALR3* in HCM. Knockout and knockdown studies in mice and zebrafish, respectively, did not demonstrate any cardiac abnormalities, suggesting that the primary functions of *CALR3* are not cardiac-related [77,78]. Additionally, there is an absence of calreticulin 3 protein expression in myocardial tissue, further questioning its involvement in direct cardiac functions [76]. Taken together, these findings argue against a significant or direct role of *CALR3* in causing HCM as a monogenic determinant. The presence of *CALR3* variants in CMP patients appears coincidental rather than causal, given the lack of segregation with the disease in family studies and the absence of functional evidence supporting a detrimental effect on cardiac structure and function.

#### 2.2.5. CDH2

The cadherin 2 gene (*CDH2*, MIM 114020) encodes N-cadherin, an essential protein in cellular adhesion in the brain and muscle tissues, including cardiac muscle. This adhesion is calcium ion-dependent and is critical for the structural integrity and function of the heart, particularly in the context of ACM [79]. Disruption in the function of N-cadherin can lead to impaired cellular cohesion and stability in the cardiac muscle, potentially triggering or exacerbating the myocardial remodeling typical of ACM [80]. Initial investigations through WES identified a variant in *CDH2*, p.(Gln229Pro), in a three-generation family affected by ARVC. In the same study, another 73 genotype-negative ARVC probands were tested, with the identification of a likely pathogenic variant in *CDH2*, p.(Asp407Asn). These variants pointed to a possible role of *CDH2* in the disease. However, these findings have not been robustly supported by additional genetic or functional studies [79]. On the other hand, immunohistochemistry studies showed a reduced content of N-cadherin in ARVC patients, suggesting a degradation of cell–cell adhesion integrity, which is crucial for normal cardiac function. These studies underscore the potential significance of *CDH2* in ACM. They also highlight the need for more extensive research to fully understand the implications of *CDH2* mutations in this condition [81].

#### 2.2.6. CSRP3

The cysteine- and glycine-rich protein 3 gene (*CSRP3,* MIM 600824) encodes a muscle LIM protein that is crucial for myogenesis, maintenance of the myocyte cytoskeleton, mechanosignaling and -transduction, and actin cytoskeleton assembly. Based on human and animal studies, *CSRP3* is implicated in both HCM and DCM [82,83]. *CSRP3* mutations were associated with both DCM and HCM in the early 2000s, with variants like p.(Cys58Gly) found in German HCM families and p.(Leu44Pro) and p.(Cys150Tyr) variants found significantly enriched in HCM cohorts. These mutations led to functional disruption in protein studies, suggesting a strong causal relationship with HCM [84,85,86]. Interestingly, the role of *CSRP3* in DCM appears to be less direct. Although knockout studies in mice show a phenotype resembling DCM and several missense mutations in DCM patients have been documented, the overall evidence pointing to a significant role of *CSRP3* in DCM is still considered limited [64,87,88,89]. Recent studies also highlight the role of synonymous variants in *CSRP3* that might influence mRNA stability, splicing, and miRNA-binding sites, which could contribute to the pathogenesis of DCM. However, a functional validation through in vitro and in vivo studies is required to assess the impact of these synonymous mutations in DCM [90]. This complex picture is further complicated by research indicating that *CSRP3* functional impact may vary significantly between species. For instance, studies using high-resolution proton magnetic resonance spectroscopy showed that *CSRP3* knockout in neonatal cardiomyocytes results in metabolic changes that could predispose to CMP, suggesting a potentially broader part in metabolic regulation within the cardiac disease [90]. Finally, whereas the evidence strongly supports a role for *CSRP3* in HCM, its involvement in DCM remains less clear, necessitating further research to clarify its mechanistic contribution and potential as a therapeutic target in CMPs.

#### 2.2.7. CTF1

The cardiotrophin 1 gene (*CTF1*, MIM 600435) encodes a member of the interleukin 6 superfamily that has been implicated in DCM pathophysiology. *CTF1* is known for its potent induction of cardiomyocyte hypertrophy and enhancement of myocyte survival, potentially implicating it in the myocardial responses observed in DCM. Despite initial genetic studies suggesting a role of *CTF1* variants in DCM, new variants, identified in studies from 2000 onwards, were found to be prevalent in the general population, generating some confusion when trying to assess a direct contribution to DCM [91,92,93]. Increased expression levels of RNA and CTF1 protein were observed in failing left ventricular myocardium, pointing to a contributory role in heart failure. Immunohistochemistry studies confirmed the elevated presence of CTF1 in cardiac myocytes from failing hearts [94]. These observations suggest that while *CTF1* is upregulated in heart failure, its direct causative link to DCM remains elusive due to the lack of conclusive functional evidence and consistent genetic segregation data.

#### 2.2.8. CTNNA3

The catenin alpha 3 gene (*CTNNA3*, MIM 607667) encodes αT-catenin, a protein crucial for cellular adhesion in cardiac tissues, particularly through its interactions with N-cadherins and desmosomal cadherins like plakophilin 2 [95]. While αT-catenin is integral to cardiac structure and function, its involvement in ACM has been difficult to establish due to conflicting evidence [14]. Initial investigations revealed a significant decrease in αT-catenin expression alongside plakophilin 2 in autopsied ARVC patients compared to controls. Furthermore, knocking down *CTNNA3* in cardiomyocytes resulted in reduced plakophilin 2 expression, mirroring ARVC’s pathological features [96]. Early research identified a rare p.(Ala689Val) variant in *CTNNA3* among Danish ARVC patients. However, this variant was not deemed disease-causing due to the lack of functional validation and segregation analysis [97]. Subsequent screening in a larger ARVC cohort identified two novel heterozygous variants within important domains of αT-catenin, which were proposed to potentially weaken interactions with β-catenin or plakoglobin. However, these findings alone were not sufficient to conclusively link *CTNNA3* mutations directly to ACM [98]. Moreover, the role of *CTNNA3* in cardiac pathology is further complicated by its involvement in multiple cellular functions beyond the desmosomal interactions. As observed in knockout mouse models, the lack of this protein regulating gap junction remodeling does not directly cause ACM, but rather leads to progressive dilated cardiomyopathy [99].

#### 2.2.9. DTNA

The dystrobrevin gene (*DTNA*, MIM 601239) encodes dystrobrevin-α, a scaffold protein for signaling molecules at the sarcolemma of cardiac muscle. This protein is involved in maintaining the structural integrity of muscle fibers by linking the extracellular matrix to the subsarcolemmal cytoskeleton [100]. *DTNA* was reported in relation to autosomal dominant DCM in a study analyzing sequence data from 7,855 CMP cases compared to the Exome Aggregation Consortium (ExAC) database, where *DTNA* variants were found in DCM patients [101]. This gene appears to associate more frequently with the left-ventricular non-compaction (LNVC, MIM 604169) phenotype. Overexpression of the LNVC-associated *DTNA* p.(Asn49Ser) variant in a mouse heart resembled a phenotype of deep trabeculation, DCM, and cardiac dysfunction [100]. There is still a lack of genetic and functional evidence to assess a possible involvement of this gene in DCM.

#### 2.2.10. EYA4

The EYA transcriptional coactivator and phosphatase 4 gene (*EYA4*, MIM 603550) encodes a factor playing a critical role in both embryonic development and inner ear function, but it has also been implicated in cardiovascular disorders, particularly in DCM [102]. This association between *EYA4* and cardiac dysfunction has begun to unfold as more genetic and functional studies are conducted. A study identified a heterozygous truncating mutation in *EYA4*, which presented with sensorineural hearing loss followed by late-onset DCM, suggesting a novel syndromic linkage between these two conditions and underscoring the importance of *EYA4* beyond the auditory system, with a potential role in cardiac function. Subsequent functional assays, including knockdown experiments in zebrafish, demonstrated that reduced EYA4 expression leads to cardiovascular dysfunction [103]. Further studies using transgenic mouse models elaborated on the mechanisms through which *EYA4* mutations could lead to DCM. For instance, overexpression of a rare *EYA4* variant in mice revealed a phenotype characteristic of DCM linked to an overexpression of p27 in cardiomyocytes, suggesting a complex regulatory role of EYA4 in cell cycle control. Additional research hypothesizes the interaction between EYA4 and the Six1 transcription factor, which together regulate crucial pathways involved in cardiac hypertrophy and cardiomyopathy [104].

#### 2.2.11. GATAD1

The GATA zinc finger domain-containing 1 gene (*GATAD1*, MIM 614518) encodes a ubiquitously expressed protein involved in all stages of embryonic mouse heart development. GATAD1 interacts with chromatin complexes for histone modification that are crucial in heart failure [105,106]. A genetic study identified a missense mutation, p.(Ser102Pro), in *GATAD1* within a consanguineous family with autosomal recessive DCM. The variant segregated with the disease, marking a significant advancement in understanding its contribution to DCM. Immunohistochemistry analysis illustrated abnormal subcellular and nuclear localization of GATAD1 in DCM-affected individuals, contrasting with its normal distribution in healthy tissues. These observations indicate the potential impact of the mutation on cellular disease mechanisms [107]. Additionally, zebrafish embryos with altered *GATAD1* expression displayed cardiac defects, emphasizing the role of the gene in heart structure and function [108].

#### 2.2.12. ILK

The integrin-linked kinase gene (*ILK*, MIM 602366) encodes a major focal adhesion protein essential for maintaining cardiomyocyte shape and ventricular morphology during embryonic development. This protein has been linked to various cardiac diseases, particularly DCM, through its role in cellular signaling and structural integrity [109]. Research has shown that mutations in *ILK*, such as p.(Ala262Val) and p.(Pro70Leu) found in DCM patients, impact cardiac function by leading to transcript destabilization and nonsense-mediated decay, particularly in model organisms like zebrafish [110]. These studies suggest a genetic predisposition to the pathogenesis of DCM linked to *ILK* mutations, which disrupt normal cellular processes and may lead to heart failure. Besides DCM, mutations in *ILK* have been recently described in patients with ACM [111]. Experimental models further underscored the therapeutic potential of *ILK* in treating heart diseases. In rat models of DCM, for instance, targeted *ILK* therapy using adenoviral vectors demonstrated a significant reduction in inflammatory cell infiltration, cardiomyocyte degeneration, and overall mortality [112]. The interactions of ILK with other proteins, such as LIM-only adaptor PINCH-1 and α-parvin, are crucial for the assembly of focal adhesions, which are complexes that facilitate cell signaling and survival through the activation of the Akt kinase pathway. This pathway is particularly significant in cardiac physiology, as it supports cell survival under stress. Mice deficient in ILK or its associated proteins exhibit severe cardiac abnormalities, underscoring the vital role of *ILK* in maintaining cardiac structure and function [113]. Moreover, the loss of ILK in murine models leads to spontaneous development of cardiomyopathy and heart failure, characterized by a significant disruption in cell adhesion and a decrease in Akt phosphorylation, crucial for cardiac stress response [114].

#### 2.2.13. JPH2

The junctophilin 2 gene (*JPH2*, MIM 605267) plays a crucial role in cardiac physiology, since its product, junctophilin 2, is essential in forming junctional complexes that bridge the plasma membrane and the endoplasmic/sarcoplasmic reticulum. This function is critical for controlling calcium signaling, which is pivotal for normal cardiac contractility and rhythm stability [115]. Functional studies underscored the involvement of *JPH2* in various cardiac pathologies, notably HCM and DCM. For instance, induced pluripotent stem cell-derived cardiomyocytes harboring *JPH2* mutations exhibit cellular hypertrophy, sarcomere disarray, and arrhythmias [116]. A founder mutation, *JPH2*:c.482C>A, p.(Thr161Lys), has been observed across several Finnish families, exhibiting autosomal dominant inheritance and co-segregation with HCM, suggesting a robust link between this particular mutation and the disease [117]. Different modes of inheritance can influence the clinical presentation and severity of the cardiac conditions associated with *JPH2* mutations. LOF mutations transmitted via an autosomal recessive pathway are linked to severe, early-onset DCM, characterized by significant cardiac dysfunction starting at a young age. Conversely, autosomal dominant missense mutations are typically associated with a broader spectrum of cardiac abnormalities, including HCM and various arrhythmias, which may also predispose to sudden cardiac death [5,118,119,120]. Genetic variants in the JPH2 gene cause distinct alterations in myocardial lipid profiles in hypertrophic and dilated cardiomyopathy mouse models, revealing unique changes in lipid composition and potential therapeutic targets [120].

#### 2.2.14. KLF10

The Krüppel-like factor 10 gene (*KLF10*, MIM 601878) plays a significant role in various biological processes, including TGF-β signaling, which influences cellular proliferation and differentiation. KLF10 is a transcription factor that has been studied primarily for its regulatory function in the immune response and its expression in response to TGF-β signaling. It acts as a potent modulator of cellular growth and has been implicated in myocardial response to stress and pathological remodeling [121]. TGF-β signaling pathways are significantly involved in the myocardial fibrotic process, contributing to the phenotypic expression of hypertrophy seen in HCM patients. This gene was linked to HCM in a cohort study that discovered six missense variants in six individuals with HCM diagnosis [121,122]. Nevertheless, the mechanisms by which *KLF10* may contribute to HCM remain to be fully delineated.

#### 2.2.15. KLHL24

The ubiquitin ligase substrate receptor Kelch-like protein 24 gene (KLHL24, MIM 611295) encodes a member of the Kelch-like protein family that acts as a substrate-specific adaptor to cullin E3 ubiquitin ligases. It ensures substrate recognition of intermediate filaments for proteasomal degradation (e.g., desmin) [123]. The first reported mutation p.(Met1Val), which leads to the first 28 N-terminal amino acid loss, caused a gain of function, i.e, excessive proteasomal degradation of the desmin protein [124,125].This was demonstrated with the use of 3D cardiac tissue engineering. Genome-wide linkage analysis and exome sequencing identified two KLHL24 homozygous mutations, p.(Glu350*) and p.(Arg306His), in two consanguineous HCM families originating from Iraq and Iran, demonstrating the gene’s involvement in CMP [122]. Endomyocardial and skeletal muscle samples from individuals of both families showed distinct changes, such as the accumulation of desmin intermediate filaments. Additionally, when the zebrafish counterpart klhl24a is reduced, it leads to heart abnormalities like those seen in other genes associated with HCM [124]. A recent study of consanguineous early-onset CMP families from Saudi Arabia reported a 14-year-old with HCM carrying a KLHL24 homozygous LOF variant, p.(Trp387*) [126]. This line of evidence supports a role for KLHL24 LOF variants in HCM, suggesting the inclusion of this gene in diagnostic panels particularly for consanguineous populations [127]. Interestingly, KLHL24 GOF variants are associated with epidermolysis bullosa simplex, a hereditary skin fragility disorder [128]. Of note, in a cohort of 20 patients with epidermolysis bullosa simplex carrying these GOF variants, 40% had DCM and were negative for screening of pathogenic variants in known DCM-associated genes [129]. This evidence provides an example of a gene with distinct LOF versus GOF effects in HCM and DCM, pointing to the diverse roles of *KLHL24* in different tissues and diseases [130].

#### 2.2.16. LAMA4

The laminin subunit alpha 4 gene (*LAMA4*, MIM 600133) is implicated in the structural integrity of the heart through the formation of laminins 8 and 9, which are major components of the basement membranes in cardiac tissues. Mutations in *LAMA4* have been associated with various cardiovascular abnormalities, including DCM. Mice deficient in *LAMA4* exhibited significant cardiovascular defects, such as endothelial disruptions, hemorrhages, and subsequent cardiac hypertrophy, leading to heart failure [131]. This phenotype mirrors the findings reported in human studies where specific mutations like p.(Pro943Leu) and p.(Arg1073*) were identified in DCM patients. These mutations, located within the integrin-interacting domain, disrupt the interaction between laminins and integrin receptors, thereby affecting the cellular adhesion and signal transduction pathways that are critical for cardiac function [110]. The profound impact of *LAMA4* on cardiovascular pathology has also been further illustrated in zebrafish models, where *LAMA4* knockdown led to severe cardiac dysfunction and hemorrhages, recapitulating some aspects of the human DCM condition [110]. Additionally, the interaction of LAMA4 with ILK, another protein implicated in cardiac disease, suggests a complex network of protein interactions that are vital for maintaining the structural and functional integrity of the heart. Mutations in both *LAMA4* and *ILK* were shown to lead to significant cardiac defects, underlying the importance of their interplay in maintaining cardiac integrity [110].

#### 2.2.17. LDB3

The LIM domain-binding 3 gene (*LDB3,* MIM 605906), encoding the protein Cypher or ZASP, plays a crucial role in cardiac and skeletal muscle structure and function. LDB3 is integral to the Z-line of sarcomeres, where it interacts with α-actinin 2 to maintain structural integrity during muscle contraction [132]. Several *LDB3* mutations were first identified in patients with late-onset DCM [133,134]. These findings were supported by animal models, where *LDB3* knockout in mice led to severe DCM and heart failure, mirroring the human disease phenotype and underscoring the gene’s functional relevance [135]. Functional investigations in zebrafish models demonstrated that *LDB3* knockdown leads to cardiac dilation and significant thinning of the ventricular walls, features typical of DCM [136]. Moreover, a recent study highlighted the severe phenotype associated with homozygous or compound heterozygous LOF variants of *LDB3*, found in multiple families [137]. In-depth functional analyses in cell models revealed that *LDB3* mutations lead to decreased cell viability, increased cardiomyocyte apoptosis, and disruptions in key signaling pathways such as Akt and p38 MAPK in cardiac models. These molecular changes provide a direct link between genetic mutations and the cascade of cellular events leading to DCM [138]. Besides DCM, mutations in *LDB3* have been recently described in patients with ACM [132].

#### 2.2.18. MT-TI

The mitochondrial tRNA isoleucine gene (*MT-TI,* MIM 590045) encodes a mitochondrial tRNA isoleucine located on the heavy stand of mitochondrial DNA. This gene was first associated with HCM, but DCM cases carrying MT-TI deletions have been also recently described [139,140,141]. In one study, the analysis of cardiac tissue from two affected HCM families segregating a homoplasmic variant m.4300A>G revealed severely decreased respiratory chain activity of mitochondria, in particular of complexes I and IV, and a decrease in mature cardiac mt-tRNAIle [140].

#### 2.2.19. MYH6

The α-cardiac myosin heavy chain gene (*MYH6,* MIM 160710) encodes the α isoform of cardiac myosin heavy chain. The α isoform is abundant in both atria and ventricles during embryogenesis. After birth, the β isoform becomes predominantly expressed [142]. The first genetic evidence of *MYH6* involvement in HCM was published in 2002, where a variant p.(Arg795Gln) was found in one HCM patient presenting symptoms after middle age [143]. This study corroborated previous functional data demonstrating that the α isoform transcripts are expressed in adult ventricular myosin (about 30%), though the abundance of αMyHC protein is very low [144]. The authors suggested that lower abundance of the α versus β isoform accounts for the late onset of HCM. Another four missense variants were then identified: one in a patient diagnosed with HCM p.(Gln1065His), and the others in three patients diagnosed with DCM—p.(Pro830Leu, p.(Ala1004Ser), and p.(Glu1457Lys) [145]. The HCM patient had an early-onset severe phenotype with death occurring in the fifth decade of life, while the three DCM patient carriers had a late-onset phenotype. All *MYH6* mutations occurred within highly conserved residues and were predicted to change either the structure or the chemical bonds of the protein. Other studies reported additional rare missense variants in DCM patients [146,147]. A study showed that degenerated myocardial cells from HCM patients had immunoreactivity for MYH6. These cells also exhibited cytoplasmic vacuolation, with vacuoles occupying a significant proportion of cell volume [148]. Functional studies on *MYH6*/DCM-associated variants showed altered myocyte contractility in ventricular myocytes from rats [149]. Notwithstanding this evidence, the relatively small amount of αMyHC protein present in healthy left ventricles has called into question the role of *MYH6* as a gene contributing to CMP [145].

#### 2.2.20. MYLK2

The light chain kinase 2 gene (*MYLK2*, MIM 606566) encodes a protein expressed in adult skeletal muscle [150]. A few mutations in *MYLK2* were reported in association with digenic forms of HCM, where patients also carried additional variants in *MYBPC3*, *MYH7*, and *FLNC* [151,152]. Unfortunately, no segregation data were available to draw any conclusions of a possible pathogenic involvement of these *MYLK2* variants. Only one sequencing study associated *MYLK2* with the DCM phenotype [101]. Moreover, there is a paucity of functional studies investigating the role of this gene in CMPs.

#### 2.2.21. MYOM1

The myomesin 1 gene (*MYOM1*, MIM 603508) encodes a protein located in the M-band, which is found in all types of vertebrate striated muscle. The N-terminal domain 1 of myomesin interacts with the a-helical tail domain of myosin, whereas myomesin domains 4–6 bind to the C-terminus of titin, stabilizing the contractile apparatus during striated muscle contraction [153]. *MYOM1* was first reported in relation to HCM in 2011 [154]. The identified p.(Val1490Ile) variant led to reduced thermal stability of the myomesin My11–13 and My12–13 dimers and a significantly decreased dimerization affinity. This variant segregated in two affected family members, although analysis of the other disease-causing genes was not performed. Additional *MYOM1* variants were identified in HCM probands, although these variants have high allele frequencies in the Genome Aggregation Database (GnomAD). Moreover, additional variants in other HCM genes were found in the same patients [155,156,157].

#### 2.2.22. MYOZ2

The myozenin 2 gene (*MYOZ2*, MIM 605602) encodes a protein important for sarcomere organization that interacts directly with crucial signaling pathways in cardiac and skeletal muscles. MYOZ2 interacts with calcineurin, a key phosphatase in calcium-dependent signaling, tethering it to alpha-actinin at the Z-line of cardiac and skeletal muscle cells via the calcium signaling [158]. The role of *MYOZ2* as a modifier gene in modulating cardiomyopathy variability has been demonstrated with a significant correlation between its expression and myocardial contractile function in mouse congenic strains [159]. Initial studies highlighted the presence of mutations in *MYOZ2*, such as p.(Ser48Pro) and p.(Ile246Met), which were found to co-segregate with HCM, suggesting a direct genetic link to the disease phenotype [160]. In a five-generation Chinese family, a missense variant in *MYH7* p.(Ala719His) co-segregated with a variant in *MYOZ2* p.(Leu169Gly). Individuals carrying both mutations displayed more severe symptoms than those with the *MYH7* mutation only, implying a modifying role of *MYOZ2* in HCM [161]. These findings are supported by mouse models demonstrating cardiac hypertrophy like the human HCM phenotype, even when calcineurin activity is not altered. This evidence indicates that the contribution of *MYOZ2* to the HCM phenotype may operate through mechanisms independent of this pathway [160,162].

#### 2.2.23. MYPN

The myopalladin gene (*MYPN,* MIM 608517) encodes a protein that plays a pivotal role in the structural organization of the sarcomere, particularly at the Z-line and I-band, interacting with essential Z-line proteins. It is increasingly clear that mutations in *MYPN* contribute to both HCM and DCM, affecting cardiac muscle integrity and function through the disruption of sarcomere and myofibrillar architecture [163,164,165]. Besides HCM and DCM, a *MYPN* nonsense mutation was also found in patients with RCM (MIM 615248) [165]. Functional studies in mice demonstrated that *MYPN* knockout showed mild cardiac dilation and systolic dysfunction, which mirrors some aspects of the DCM phenotype in humans [166]. Moreover, the broader impact of *MYPN* mutations has been noted in diverse populations. A study involving Lebanese and Chinese cohorts highlighted the presence of *MYPN* variations in DCM patients, further emphasizing the role of this gene in CMPs across different ethnicities [167,168].

#### 2.2.24. NEBL

*NEBL* (MIM 605491) encodes the cardiac-specific isoform nebulette, a crucial member of the nebulin family, predominantly involved in the structural integrity of the sarcomere within heart muscle. Highly expressed in cardiac muscle, nebulette binds actin, interacts with thin filaments, and associates with Z-line proteins in striated muscle, playing an essential role in cardiac myofibril assembly. It acts as a pivotal link between sarcomere actin and desmin intermediate filaments within the heart muscle sarcomeres, being integral for maintaining heart muscle integrity and functionality [169]. Mutations in *NEBL* were associated with various cardiac pathologies, including DCM [170]. Functional studies using transgenic mice revealed that *NEBL* mutations can significantly affect sarcomere ultrastructure, disrupt cellular contractile function, and impair calcium homeostasis, underscoring the gene role in DCM pathogenesis [171,172]. For instance, mutations such as p.(Lys60Asn), p.(Gln128Arg), p.(Gly202Arg), and p.(Ala592Glu) were observed to cause various alterations in cardiac structure and function, ranging from embryonic lethality to progressive heart failure in adulthood [172]. Moreover, a knockout mouse model for *NEBL* exhibited widened Z-lines and increased expression of cardiac stress markers, suggesting that disease-causing mutations in *NEBL* likely exert dominant GOF effects [159]. Diversity in phenotypic outcomes was observed across 7 patients in a cohort of 389 patients affected by DCM, HCM and LVNC: four missense mutations were identified in DCM and HCM patients and one in a LVNC patient. While HCM and DCM related mutations were within the nebulin-like repeats, responsible for actin binding, the associated LVNC mutation was in the C-terminal serine-rich linker region. This evidence suggests that specific locations and nature of *NEBL* mutations can variably influence cardiac muscle function, leading to a distinct pathological cardiac phenotype [173].

#### 2.2.25. NEXN

The nexilin gene (*NEXN,* MIM 613121) encodes a protein critical for the structural and functional integrity of cardiac and skeletal muscles. NEXN is involved in regulating the actin cytoskeleton and sarcomere assembly, key components in maintaining cardiomyocyte stability and responding to mechanical stress [174]. Mutations in *NEXN* were associated with a spectrum of cardiac pathologies, including DCM, HCM, and sudden cardiac death [175,176,177]. Disruption of nexilin can lead to severe cardiac abnormalities. Constitutive knockout models in mice demonstrated that the absence of *NEXN* leads to rapid progression of cardiomyopathy, characterized by left ventricular dilation, thinning of the ventricular walls, and a decline in cardiac function [175,178]. Genetic investigations revealed the involvement of *NEXN* in cardiac diseases. For example, a *NEXN* variant p.(Glu575*) was found together with a novel *SCN5A* variant in a family with progressive DCM and cardiac arrhythmias, suggesting a synergistic effect of the two mutations on disease pathogenesis [179]. Targeted NGS of 102 genes in Han Chinese patients with idiopathic DCM revealed *TTN* truncating variants as predominant, followed by variants in *LMNA*, *RBM20*, and *NEXN*, providing molecular diagnosis in 34.7% of patients and highlighting insights into genotype–phenotype correlations [180]. Pediatric and fetal cases further emphasized the impact of *NEXN* mutations, with conditions ranging from transient to severe DCM and cardiomegaly in patients carrying heterozygous and homozygous mutations [181,182]. Moreover, an evaluation of *NEXN* variants in patients with cardiomyopathy or sudden cardiac death showed a predominance of DCM, with particularly severe and early-onset phenotypes in those with double *NEXN* variants [183]. Pathogenic variants were also detected in patients with HCM and sudden infant death syndrome/idiopathic ventricular fibrillation, further complicating the phenotypic spectrum of *NEXN* mutations.

#### 2.2.26. NKX2-5

The NK2 homeobox 5 (*NKX2-5,* MIM 600584) gene encodes a homeobox transcription factor essential for heart development and function, specifically in regulating genes involved in cardiac morphogenesis, including sarcomere organization and the development of the conduction system [184,185,186]. Mutations in *NKX2-5* were linked to various congenital heart diseases, such as atrial septal defects, ventricular septal defects, and anomalies in the conduction system [187,188]. The critical role of *NKX2-5* in cardiac health was highlighted in 2013, marking the first association of its mutations with adult-onset DCM. These mutations were suggested to contribute to DCM by altering protein degradation and transcriptional activity, establishing a new avenue for understanding the molecular basis of this disease [189]. Subsequent research in 2014 identified a novel heterozygous mutation, p.(Ser146Trp), in a family displaying an autosomal dominant DCM pattern. This mutation was found among 130 unrelated patients with idiopathic DCM and demonstrated complete penetrance, with the DCM phenotype also associated with arrhythmias. Functional analyses indicated that this variant resulted in significantly reduced transcriptional activity compared to the wild-type protein, suggesting a mechanism through which *NKX2-5* mutations disrupt heart function [190]. Further studies identified additional variants in *NKX2-5* among patients with sporadic adult-onset DCM. These mutations lead to reduced transcriptional activity and disrupted interactions with key cardiac transcription factors. This evidence supports a model where *NKX2-5* mutations contribute to DCM pathogenesis through altered gene regulation and impaired transcriptional networks, highlighting their significant impact on cardiac structural and functional integrity [191,192].

#### 2.2.27. OBSCN

The obscurin gene (*OBSCN*, MIM 608616) encodes a giant sarcomere signaling protein crucial for myofibrillogenesis, cytoskeletal organization, and cell adhesion. These characteristics facilitate the interactions between the sarcoplasmic reticulum and myofibrils, which are essential for the structural integrity and function of muscle cells. The physiological roles of obscurin were initially elucidated through a knockout mouse model, which revealed significant implications in skeletal muscle but less understood effects in cardiac muscle [193,194,195,196]. *OBSCN* mutations were associated with both HCM and DCM. In HCM, the p.(Arg4344Gln) and p.(Ala4484Thr) variants affect the binding of obscurin with titin domains, which is crucial for the structural organization of the sarcomere. In particular, the p.(Arg4344Gln) mutation was shown to impair obscurin localization to the Z-band, suggesting a potential mechanism by which this mutation contributes to the pathogenesis of HCM [197]. Recent studies also implicated *OBSCN* mutations in DCM. A study identified five potentially disease-causing *OBSCN* mutations in patients with familial DCM, suggesting a significant role for obscurin, potentially through mechanisms like haploinsufficiency which affects protein expression levels and disrupts normal cardiac function [198].

#### 2.2.28. PDLIM3

The PDZ and LIM domain 3 gene (*PDLIM3*, MIM 605889) encodes a protein with both a PDZ domain and a LIM domain, suggesting its potential role in cytoskeletal organization. This protein has been shown to bind to the spectrin-like repeats of alpha-actinin 2 and to co-localize with alpha-actinin 2 at the Z lines in skeletal muscle [199]. This gene is also referred to as *ALP*. Research involving *ALP*-knockout mice revealed alterations in regional systolic function and hypertrophy, primarily due to its specific expression in the right ventricular outflow tract. These changes suggest a reduction in contractile function and an increase in wall thickness as a response to chronic hypoxia [200]. Variants in *PDLIM3* were identified in two unrelated HCM patients. In the specific, a non-synonymous VUS p.(Glu106Ala) was found in one HCM patient [201]. A large deletion involving the first four exons of *PDLIM3* was identified in a patient from a cohort of 505 unrelated HCM patients screened for copy number variations (CNVs) [202]. Functional characterization of these variants in HCM phenotype are yet to be performed to further investigate the involvement of this gene in CMPs.

#### 2.2.29. PLEKHM2

The pleckstrin homology and RUN domain-containing M2 gene (*PLEKHM2,* MIM 609613) encodes a protein predominantly involved in lysosomal trafficking [203,204,205]. In a large Bedouin family with severe recessive DCM and LVNC, WES revealed a novel *PLEKHM2*:c.2156_2157delAG variant, causing a frameshift and exon skipping. PLEKHM2 regulates endosomal trafficking, and this mutation led to abnormal subcellular distribution of endosomes and lysosomes, as well as impaired autophagy flux in patients’ fibroblasts. Transfection with wild-type *PLEKHM2* cDNA restored lysosomal distribution, implicating *PLEKHM2* in DCM and LVNC pathogenesis via autophagy disruption [206].

#### 2.2.30. PRDM16

The PR domain-containing 16 gene (*PRDM16*, MIM 605557) encodes a zinc-finger-containing transcription factor that promotes or represses tissue-specific gene expression [207]. PRDM16 is expressed in mouse and human hearts [208,209]. Missense variants in *PRDM16* were primarily associated with DCM, whereas nonsense and frameshift mutations were associated with LVNC [210]. Subsequently, *PRDM16* has also been associated with pediatric DCM, a disorder that typically remains clinically silent until adulthood. Genetic screening in families with pediatric DCM cases identified a de novo frameshift *PRDM16* variant in a proband diagnosed at four months of age [211]. This variant resulted in the addition of an anomalous peptide tail consisting of 48 residues and prematurely truncating the protein product. *PRDM16* is predicted to be highly intolerant to LOF. In zebrafish, *PRDM16* was shown to have a dominant-positive effect on cardiomyocyte proliferation, with overexpression or knockdown resulting in impaired cardiomyocyte proliferation [210]. On the other hand, functional studies linked this gene to HCM. In fact, *PRDM16*-knockout mice develop cardiac hypertrophy, excessive fibrosis, and mitochondrial and metabolic defects, leading to heart failure as they age [207]. Further studies are therefore essential to assess *PRDM16* role in CMPs.

#### 2.2.31. PSEN2

The presenilin 2 gene (*PSEN2,* MIM 600759) encodes a transmembrane key protein implicated in Alzheimer’s disease (AD), particularly in the inherited forms of the disease. Mutations in *PSEN2*, along with mutations in *PSEN1* and amyloid precursor protein (*APP*), are known to cause familial AD. PSEN2, along with PSEN1, plays a critical role in regulating the processing of APP through its involvement in gamma-secretase, an enzyme responsible for cleaving APP and the cleavage of the Notch receptor. *PSEN1* and *PSEN2* are expressed in the heart with a potential role in cardiac development. Studies conducted in 132 families with DCM and 183 individuals with idiopathic DCM revealed a novel *PSEN1* mutation in one family and a single *PSEN2* missense mutation in two other families. Importantly, both mutations were found to be present in all clinically affected individuals and showed segregation with DCM and heart failure within these families. Cultured skin fibroblasts from individuals carrying mutations in *PSEN1* and *PSEN2* exhibited changes in calcium signaling [212]. *PSEN2*-knockout mice displayed normal cardiac development without hypertrophy or fibrosis. Notably, they exhibited increased cardiac contractility, supported by higher Ca^2+^ transients and tension in isolated papillary muscles. *PSEN2* was found to interact with sorcin and ryanodine receptor 2, both crucial for cardiac function, indicating a role of this gene in excitation-contraction coupling [213].

#### 2.2.32. RPS6KB1

This gene (MIM 608938) encodes a protein (ribosomal protein S6 kinase beta 1 or S6K1) belonging to the ribosomal S6 kinase family, which comprises serine/threonine kinases. It is activated by mTOR signaling, facilitating protein synthesis, cell growth, and proliferation [214]. Exome and targeted sequencing of 401 Indian HCM patients revealed a novel heterozygous missense variant in *RPS6KB1* in two unrelated families p.(Glu47Trp), co-segregating with the phenotype. Replication study in a UK Biobank cardiomyopathy cohort (*n* = 190) identified other two *RPS6KB1* heterozygotes variants: p.(Gln49Lys) and p.(Tyr62His). Functional analysis showed that mutant proteins activated signaling cascades, suggesting a GOF effect. Additionally, the same authors observed a *RPS6KB1* variant p.(Pro445Ser) in an HCM patient from Saudi Arabia [215].

#### 2.2.33. RYR2

The ryanodine receptor 2 gene (*RYR2*, MIM 180902) covers 107 exons and encodes a protein involved in calcium signaling in cardiac muscle cells. It is predominantly found in the sarcoplasmic reticulum (SR). RYR2 plays a crucial role in regulating the release of calcium ions from the sarcoplasmic reticulum into the cytoplasm of cardiac muscle cells during excitation–contraction coupling, a process essential for cardiac muscle contraction [216,217,218]. A functional link between RyR2 and cMyBP-C (encoded by *MYBPC3*) is known, and it suggests a potential mechanistic association between cytosolic soluble cMyBP-C and SR-triggered Ca^2+^ release through RyR2. This interaction could be clinically significant in CMPs such as HCM [219]. *RYR2* was first associated with HCM when a missense mutation p.(Thr1107Met) was identified in an affected proband, co-segregating with asymmetric septal hypertrophy in the family members [220]. Functional evidence showed that the p.(Ala1107Met) mutation raises the level at which calcium release stops and reduces the proportion of released calcium [221]. Three patients with apical hypertrophy carried two missense mutations (p.(Glu3809Gly) and p(.Arg929His)) and one splicing mutation (c.7966-2A>T) in *RYR2*. The mutations were predicted to be damaging and affecting highly conserved residues, possibly contributing to HCM pathogenesis [222]. Another novel mutation p.(Pro1124Leu) was identified in an HCM patient. Mice homozygous for the variant showed a clinical phenotype like a human patient, with overexpression of calmodulin as a potential hypertrophic mediator [218,223]. A recent study performed in an Indian cohort of 22 patients with CMPs identified three patients as carriers of *RYR2* variants of uncertain significance [224]. In addition, mutations in *RYR2* have been identified in ACM patients [225].

#### 2.2.34. SGCD

The *SGCD* gene (MIM 601411) encodes the δ-sarcoglycan protein. This protein is part of the sarcoglycan complex, consisting of four components, within the larger dystrophin-glycoprotein complex (DGC) that connects the F-actin cytoskeleton to the extracellular matrix. Its highest expression occurs in skeletal and cardiac muscle, and it is involved in maintaining the structural integrity of muscle fibers [226,227]. This gene was first associated with autosomal recessive limb girdle muscular dystrophy and then with DCM, when a missense variant and a deletion were identified in three DCM patients [228]. Subsequently, the p.(Arg71Thr) variant was identified in two members of a small DCM family in Finland [229]. Functional studies reveal that this mutation exhibits a dominant negative effect on the function of the dystrophin–glycoprotein complex, resulting in myocardial mechanical instability that can contribute to DCM development [230].

#### 2.2.35. TBX20

The T-box transcription factor 20 gene (*TBX20,* MIM 606061) encodes a member of the T-box family of transcription factors. These factors play crucial roles in the regulation of developmental processes. TBX20 is particularly involved in heart development, since it regulates the expression of genes essential for cardiac morphogenesis and function. *TBX20* mutations were associated with various congenital heart diseases, highlighting its relevance in cardiac development [231,232,233,234]. The association between *TBX20* and DCM was initially documented in 2007 [231]. A novel heterozygous mutation in *TBX20* p.(Phe256Ile) was found in a family with autosomal dominant DCM, showing complete penetrance. This variant impaired TBX20 transcriptional activity and reduced the synergistic activation of TBX20 with NKX2-5 or GATA4 [235]. Endomyocardial biopsies obtained from idiopathic DCM patients showed elevated *TBX20* mRNA levels, similarly to what reported in DCM rats [236]. A recent study investigated the association between *TBX20* truncating variants (*TBX20*tv) and DCM or LVNC. *TBX20*tv was found to be enriched in DCM/LVNC cases compared to controls, with strong co-segregation and a non-aggressive clinical phenotype characterized by low incidence of major cardiovascular events [237].

#### 2.2.36. TCAP

This gene (MIM 604488) spans two exons and encodes a protein (titin-cap) found in both striated and cardiac muscle that binds to the titin Z1 and Z2 domains. Titin-cap is a substrate of titin kinase, thought to be critical for sarcomere assembly [238]. *TCAP* mutations were identified in a cohort of 346 HCM and 136 DCM patients. Functional assays revealed that HCM-linked mutations enhanced titin-cap interaction with titin and calsarcin 1, whereas DCM-linked mutations impaired interactions with muscle LIM protein, titin, and calsarcin 1. This evidence suggested a correlation between mutation effects and clinical phenotypes [239]. Among a cohort of 30 unrelated DCM patients, one novel variant was found in a patient. However, no family data were available for the segregation analysis [240]. In 2021, a study of 230 unrelated DCM patients of Vietnamese origin identified the p.(Arg158Cys) variant, classified as likely pathogenic according to the ACMG [241].

#### 2.2.37. TGFB3

*TGFB3* (MIM 190230) encodes a protein called transforming growth factor beta 3 (TGF-β3), which is a member of the TGF-β superfamily. TGF-β3 is involved in various cellular processes, including cell growth, differentiation, apoptosis, and immune regulation. It plays a critical role in embryonic development, tissue repair, and wound healing [242]. This gene was first associated with ACM in 1994. The study was conducted when linkage was established with markers on chromosome 14q42.3 within a sizable Italian family, with a critical interval containing 40 known genes. Subsequently, this region was linked to various other Italian families. Sequence analysis of candidate genes within this region was conducted [243,244,245]. A nucleotide substitution (c.-36G>A) was identified in the 5’ UTR of *TGFB3*, consistently associated with the typical ARVC clinical features in affected family members, as per established diagnostic criteria. Further examination involving 30 unrelated ARVC patients conducted using denaturing high-performance liquid chromatography revealed an additional mutation (c.1723C>T) in the 3’ UTR of one proband [246]. In another study, two coding variants of uncertain significance were detected in two ARVC probands [247]. Functional studies are critical to further establish a possible pathogenic involvement of this gene in ACM pathophysiology.

#### 2.2.38. TJP1

The zonula occludens 1 gene (*TJP1*, MIM 601009) encodes a scaffolding protein localized at the intercalated discs of cardiomyocytes. It interacts with key cardiac proteins like connexin 43, N-cadherin, αT-catenin, and actin, playing a significant role in cardiac function and structure [248,249]. Initial identification of *TJP1* mutations in ACM was reported in 2018, marking a significant advancement in the understanding of ACM pathogenesis. Patient cohorts negative for mutations in known ACM-associated genes were analyzed with WES, leading to the identification of *TJP1* variants, such as p.(Arg265Trp) in an Italian ARVC cohort and p.(Ser329Leu) and p.(Asp360Val) in a Dutch–German ACM cohort. These variants, particularly p.(Tyr669Cys), are located in critical regions of the gene and are predicted to be deleterious [248,249]. Molecular dynamic simulations suggested that these mutations could significantly impact the structural stability and functional interaction of the protein within the cardiac tissue [250].

#### 2.2.39. TNNI3K

The troponin I-interacting kinase protein was first identified as a cardiac-specific protein kinase that interacts with cardiac troponin I [251]. The overall domain structure of TNNI3K resembles that of ILK (see above), suggesting similar functions. TNNI3K is suggested to probably interact with additional sarcomere proteins such as cardiac α-actin and myosin binding protein C, proposing a role for TNNI3K in the modulation of sarcomere function through interactions with key components of the sarcomere complex [252]. The first association of *TNNI3K* (MIM 613932) with DCM was reported in a study of a multigenerational family with a particular cardiac phenotype characterized by variably expressed atrial tachyarrhythmia, conduction system disease, and DCM [253]. Linkage analysis and WES were used to identify a novel *TNNI3K* variant that resulted in abnormal peptide aggregation in vitro. Of note, ventricular tissue obtained from a mutation carrier displayed histopathological features of DCM. In parallel, reduced TNNI3K protein staining with unique amorphous nuclear and sarcoplasmic inclusions was observed. Other two important studies reported *TNNI3K* novel variants in families with DCM phenotype, segregating with the disease [252,254]. In one study, conducted on three different families with phenotype predominantly consisting of supraventricular tachycardia occurring together with cardiac conduction disease and/or DCM, the same *TNNI3K* variant was identified, p.(Glu768Lys). It co-segregated with disease features in all affected individuals (*n* = 23) from all three families [254]. This variant displayed enhanced kinase activity consistent with previous mouse studies that demonstrated increased conduction indices and cardiomyopathy development with increased TNNI3K levels [252]. In another study, a splice site variant (c.333+2 T>C) was identified and co-segregated with the disease in affected family members. The variant was predicted to result in a premature stop codon falling in exon 4 and to induce nonsense-mediated mRNA decay. Real-time qPCR also confirmed that *TNNI3K* mRNA expression decreased significantly compared with the controls [255]. Lastly, an increased burden of rare coding *TNNI3K* variants in DCM patients was reported, with two additional new likely pathogenic *TNNI3K* variants associated with increased autophosphorylation, suggesting that enhanced autophosphorylation might be the pathogenic mechanism caused by *TNNI3K* variants [256].

#### 2.2.40. TRIM63

The tripartite motif-containing protein 63 gene (*TRIM63*, MIM 606131) encodes the E3 ubiquitin–protein ligase, also known as muscle-specific RING finger protein 1, which is involved in the degradation of sarcomere proteins through ubiquitylation [257]. This protein is crucial in maintaining cardiac muscle integrity and function. Homozygous or compound heterozygous variants in *TRIM63* were associated with HCM. Evaluations of affected families showed that only homozygous and compound heterozygous carriers exhibited signs of the disease, while heterozygous carriers remained healthy. A study proposed a modifier role for heterozygous missense variants in patients with HCM [258]. Consistent with an LOF mechanism for *TRIM63* variants, mice with double knockouts for both *TRIM63* and *TRIM55* exhibited severe cardiac hypertrophy [36,259]. One study found that 15 cases of HCM were linked to rare homozygous or compound heterozygous variants in *TRIM63*. These cases often presented with left ventricular hypertrophy, and in some instances left ventricular systolic dysfunction. Additionally, these patients frequently exhibited non-sustained ventricular tachycardia, and some suffered adverse cerebrovascular events [36]. Further studies confirmed *TRIM63* as an uncommon cause of HCM inherited in an autosomal recessive manner, associated with significant cardiac manifestations including concentric left ventricular hypertrophy and a high rate of left ventricular dysfunction [260].

#### 2.2.41. VCL

The vinculin gene (*VCL*, MIM 193065) encodes a mechanosensitive protein incorporated in Z-disks, of which two splicing isoforms are produced. Metavinculin is 68 residues longer than vinculin, with exon 19 additionally spliced in. Both isoforms are expressed in the human heart. Few studies have reported variants in *VCL* associated with DCM and HCM [261,262]. One study sequenced *VCL* exon 19 only in 350 DCM patients and detected two *VCL* variants (one missense and one in-frame deletion) in three patients. Viscometry analysis and electron microscopy demonstrated that both variants altered actin filament organization [263]. In addition, the missense variant also causes disruption of the intercalated disks. The missense variant segregated in a DCM family together with another candidate variant in *MYBPC3*. In 2020, a large cohort-based study performed in 2538 patients highlighted a significant enrichment of predicted protein-truncating and missense *VCL* variants compared with the ExAC population database [264]. A second large cohort-based study demonstrated again significant enrichment of rare, predicted protein-truncating *VCL* variants in DCM patients [265]. Functional studies were performed in mice, zebrafish, and Drosophila, and they supported a role of *VCL* in myocardial function pointing to LOF as a disease mechanism [266,267,268,269,270,271,272]. *VCL* variants were associated with HCM. The latter finding is based on the role of VCL in maintaining cellular structure and transmitting mechanical forces necessary for normal heart function. In fact, when these functions are disrupted by mutations, *VCL* may contribute to HCM. Reduced VCL protein expression is lethal in germline homozygous knockout mice and leads to stress-induced HCM in heterozygotes. An induced pluripotent stem cell line was recently generated from a DCM patient carrying a heterozygous *VCL* variant and was differentiated into cardiomyocytes. The latter might serve as a disease model to better understand the molecular mechanisms and pathogenesis of *VCL* in DCM [273].

## 3. Gene–Disease Validity and Clinical Evidence

Based on the evaluated sources (ClinGen, *GeneReviews*, EMQN, and expert consensus statement), the genes listed in Table 1 are considered definitive for their respective pathological conditions. Table 2 provides an overview of the minor genes and highlights how different sources classify genes in relation to their association with specific genetic diseases. Despite some discrepancies, there is generally a consensus on assessing the definitive role of these genes, supported by current evidence. This joint evaluation from various sources provides a clear and updated view of gene–disease validity for CMPs. In Table 3, the counts of VUS and P/LP variants reported in ClinVar for these minor genes are detailed (entry May 2024) and relevant animal studies found in PubMed are reported. Notably, some genes, such as *KLF10* and *OBSCN*, have very few recorded entries, while other genes like *RPS6KB1* and *TJP1* have no entries at all. In Figure 2, the charts show the distribution of variants for ACM, DCM, and HCM. In ACM, *CTNNA3* has the most variants, predominantly VUS. *CDH2* and *TGFB3* show a notable number of pathogenic variants. In DCM, *ABCC9* and *MYH6* have a high number of variants, predominantly VUS. *MYPN* and *TCAP* have a significant number of pathogenic variants. HCM genes like *MYH6* and *MYOM1* have the most variants, with a predominance of VUS. Genes such as *MYPN* show a significant number of pathogenic variants.

## 4. Conclusions

CMPs are genetically heterogeneous diseases, meaning that they can be caused by mutations in multiple genes (such as double mutations or polygenic risk score) [275,276], some of which may have minor or additive effects. The minor genes discussed in this review may not be the primary drivers of disease, such as *MYH7* and *TNNT2* in HCM or *DSP* and *PKP2* in ACM, but their involvement may contribute to the clinical phenotype and influence the prognosis. For instance, a mutation in a minor gene might not cause a clinically observable effect when present alone, but when combined with another pathogenic variant can exert a synergistic effect, leading to a more severe form of cardiomyopathy [17,277,278]. For this reason, we felt the need to report all current knowledge on these less understood genetic factors involved in CMPs. Future studies are needed to clearly assess their contribution. Furthermore, understanding the role of minor genes is critical for providing important insights that can lead to innovative therapies. For instance, research has demonstrated that using adenoassociated virus serotype 9 (AAV9) to deliver functional PKP2 into Pkp2-deficient mouse models can improve heart function and survival, offering a promising therapeutic approach [279,280].

In conclusion, whereas current guidelines provide a well-established framework for genetic testing in inherited CMPs and arrhythmias, the increasing knowledge on minor genes may lead to their practical use in clinical applications.

## Figures and Tables

**Figure 1 ijms-25-09787-f001:**
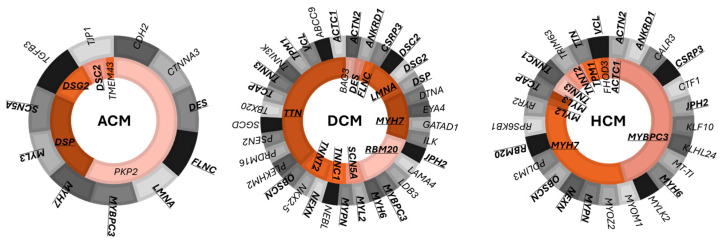
Representation of genes associated with ACM, DCM, and HCM. The circle sections are color-coded, with orange nuances indicating definitive classification and gray nuances indicating minor (moderate and/or limited) classification. Underlined genes are implicated in more conditions.

**Figure 2 ijms-25-09787-f002:**
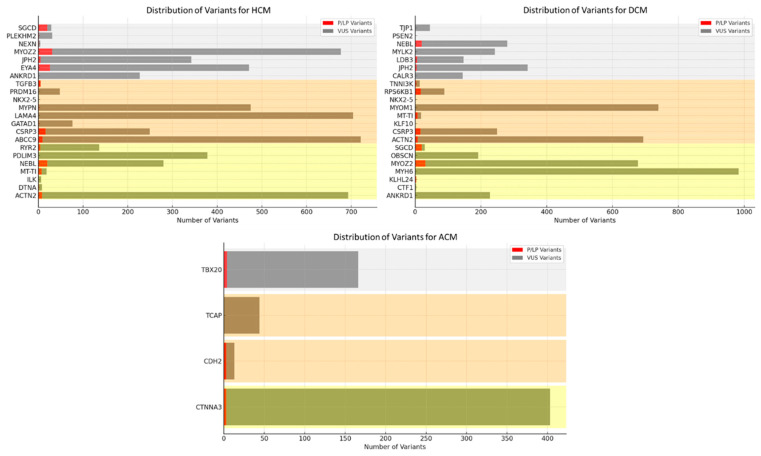
Distribution of VUS and P/LP variants in minor genes reported in ClinVar [47] in ACM, DCM, and HCM cases. The bars are color-coded, with red indicating P/LP variants and gray indicating variants of uncertain significance. The background colors represent the ClinGen [45,47] classification for each gene: orange for moderate, yellow for limited, and light gray for not associated.

**Table 1 ijms-25-09787-t001:** List of genes definitively associated with ACM (ARVC, ALVC, and/or BiVACM), DCM, and HCM according to: ClinGen [45], *GeneReviews* [42], EMQN 2023 [43,44], and a 2022 expert consensus statement [53]. *ALPK3*, *JUP*, and *PLN* are not included.

Gene	Location (GRCh38)	Hereditary	Definitive Phenotype Association [45]	N. P/LP Variants	N. VUS	Other Phenotype Associations			
						ClinGen [45]	*GeneReviews* [42]	EMQN [43,44]	Expert Consensus Statement [53]
*ACTC1*	15q14	*AD*	HCM	8	290	DCM (moderate)	DCM (moderate)	DCM (moderate)	DCM (moderate)
*BAG3*	10q26.11	*AD*	DCM	101	513	-	-	-	-
*DES*	2q35	*AD*	DCM	16	58	ACM (moderate)	ACM (moderate)	ACM (moderate)	-
*DSC2*	18q12.1	*AD, AR*	ACM (ARVC, less frequent ALVC/BiVACM)	57	667	DCM (moderate)	-	-	-
*DSG2*	18q12.1	*AD*	ACM (ARVC, ALVC, BiVACM)	85	753	DCM (limited)	DCM (limited)	-	-
*DSP*	6p24.3	*AD*	ACM (ALVC/BiVACM, occasional hair/skin features)	472	1430	-	DCM (definitive)	DCM (definitive)	DCM (definitive)
*FHOD3*	18q12.2	*-*	HCM	4	24	-	-	-	-
*FLNC*	7q32.1	*AD*	DCM	359	1697	-	-	ACM (limited)	-
*LMNA*	1q22	*AD*	DCM	86	229	ACM (limited)	-	-	-
*MYBPC3*	11p11.2	*AD*	HCM	706	1228	ACM; DCM (limited)	DCM (limited)	-	-
*MYH7*	14q11.2	*AD*	DCM; HCM	291	1642	ACM (limited)	-	-	-
*MYL2*	12q24.11	*AD*	HCM	12	206	DCM (limited)	DCM (limited)	-	-
*MYL3*	3p21.31	*AD*, *AR*	HCM	3	197	ACM (limited)	-	-	-
*PKP2*	12p11.21	*AD*	ACM (ARVC, less frequent ALVC/BiVACM)	271	701	-	-	-	-
*RBM20*	10q25.2	*AD*	DCM	16	676	HCM (limited)	-	-	-
*SCN5A*	3p22.2	*AD*	DCM	19	252	ACM (limited)	-	-	-
*TMEM43*	3p25.1	*AD*	ACM (ARVC, BiVACM)	2	378	-	-	-	-
*TNNC1*	3p21.1	*AD*	DCM	7	139	HCM (definitive)	HCM (moderate)	HCM (moderate)	HCM (moderate)
*TNNI3*	19q13.42	*AD*	HCM	41	247	DCM (moderate)	DCM (moderate)	DCM (moderate)	DCM (moderate)
*TNNT2*	1q32.1	*AD*	DCM; HCM	53	281	-	-	-	-
*TPM1*	15q22.2	*AD*	HCM	24	267	DCM (moderate)	DCM (moderate)	DCM (moderate)	DCM (moderate)
*TTN*	2q31.2	*AD*	DCM	2381	5144	ACM; HCM (limited)	HCM (limited)	-	-

**Table 2 ijms-25-09787-t002:** Minor genes discussed in this review. They were categorized as moderate, limited, not associated, or not curated based on ClinGen terminology and their association with different forms of CMPs. Definitive genes are not included. AD = autosomal dominant, AR = autosomal recessive, SD = autosomal semidominant, - = gene not curated by source.

			ClinGen [45]	*GeneReviews* [42]	EMQN [43,44]	Expert Consensus Statement [53]	ClinGen [45]	*GeneReviews* [42]	EMQN [43,44]	Expert Consensus Statement [53]	ClinGen [45]	*GeneReviews* [42]	EMQN [43,44]	Expert Consensus Statement [53]
Gene	Location (GRCh38)	Hereditary	ACM				DCM				HCM			
*ABCC9*	12p12.1	*AD*	not associated	not associated	-	not associated	limited	limited	-	not associated	not associated	not associated	-	not associated
*ACTN2*	1q43	*AD*	not associated	not associated	not associated	not associated	not associated	moderate	moderate	moderate	not associated	moderate	not associated	moderate
*ANKRD1*	10q23.31	*AD*	not associated	not associated	-	-	limited	limited	-	-	not associated	limited	-	-
*CALR3*	19p13.11	*AD*	not associated	not associated	-	-	not associated	not associated	-	-	not associated	limited	-	-
*CDH2*	18q12.1	*AD*	limited	not associated	-	not associated	not associated	not associated	-	not associated	not associated	not associated	-	not associated
*CSRP3*	11p15.1	*AD, SD*	not associated	not associated	-	not associated	limited	limited	-	not associated	moderate	moderate	-	moderate
*CTF1*	16p11.2	*AD*	not associated	not associated	-	-	limited	limited	-	-	not associated	not associated	-	-
*CTNNA3*	10q21.3	*AD*	limited	not associated	-	not associated	not associated	not associated	-	not associated	not associated	not associated	-	moderate
*DTNA*	18q12.1	*AD*	not associated	not associated	-	-	limited	limited	-	-	not associated	not associated	-	-
*EYA4*	6q23.2	*AD*	not associated	not associated	-	-	limited	limited	-	-	not associated	not associated	-	-
*GATAD1*	7q21.2	*AR*	not associated	not associated	-	-	limited	limited	-	-	not associated	not associated	-	-
*ILK*	11p15.4	*AD*	not associated	not associated	-	-	limited	limited	-	-	not associated	not associated	-	-
*JPH2*	20q13.12	*AD, SD*	not associated	not associated	not associated	not associated	moderate	moderate	moderate	moderate	moderate	moderate	not associated	moderate
*KLF10*	8q22.3	*AD*	not associated	not associated	-	-	not associated	not associated	-	-	limited	limited	-	-
*KLHL24*	3q27.1	*AR*	not associated	-	-	-	not associated	-	-	-	moderate	-	-	-
*LAMA4*	6q21	*AD*	not associated	not associated	-	-	limited	limited	-	-	not associated	not associated	-	-
*LDB3*	10q23.2	*AD*	not associated	not associated	-	not associated	limited	limited	-	not associated	not associated	not associated	-	not associated
*MT-TI*	-	*-*	not associated	-	-	-	not associated	-	-	-	moderate	-	-	-
*MYH6*	14q11.2	*AD*	not associated	not associated	-	-	limited	limited	-	-	limited	limited	-	-
*MYLK2*	20q11.21	*AD*	not associated	not associated	-	-	not associated	not associated	-	-	not associated	limited	-	-
*MYOM1*	18p11.31	*AD*	not associated	not associated	-	-	not associated	not associated	-	-	not associated	limited	-	-
*MYOZ2*	4q26	*AD*	not associated	not associated	-	-	not associated	not associated	-	-	not associated	limited	-	-
*MYPN*	10q21.3	*AD*	not associated	not associated	-	-	not associated	limited	-	-	not associated	limited	-	-
*NEBL*	10p12.31	*AD*	not associated	not associated	-	-	limited	limited	-	--	not associated	not associated	-	-
*NEXN*	1p31.1	*AD*	not associated	not associated	-	-	moderate	moderate	-	-	limited	limited	-	-
*NKX2-5*	5q35.1	*AD*	not associated	not associated	-	-	limited	limited	-		not associated	not associated	-	-
*OBSCN*	1q42.13	*AD*	not associated	not associated		-	limited	limited	-	-	limited	not associated	-	-
*PDLIM3*	4q35.1	*AD*	not associated	not associated	-	-	not associated	not associated	-	-	limited	limited	-	-
*PLEKHM2*	1p36.21	*AR*	not associated	not associated	-	-	limited	limited	-	-	not associated	not associated	-	-
*PRDM16*	1p36.32	*AD*	not associated	not associated	-		limited	limited	-	-	not associated	not associated	-	-
*PSEN2*	1q42.13	*AD*	not associated	not associated	-	-	limited	limited	-	-	not associated	not associated	-	-
*RPS6KB1*	17q23.1	*AD*	not associated	not associated	-	-	not associated	not associated	-	-	limited	not associated	-	-
*RYR2*	1q43	*AD*	not associated	not associated	-	-	not associated	not associated	-	-	limited	limited	-	-
*SGCD*	5q33.2-q33.3	*AD*	not associated	not associated	-	-	limited	limited	-		not associated	not associated	-	-
*TBX20*	7p14.2	*AD*	not associated	not associated	-	-	limited	limited	-	-	not associated	not associated	-	-
*TCAP*	17q12	*AD*	not associated	not associated	-	-	limited	limited	-	-	not associated	limited	-	-
*TGFB3*	14q24.3	*AD*	limited	not associated	-	-	not associated	not associated	-	-	not associated	not associated	-	-
*TJP1*	15q13.1	*AD*	limited	not associated	-	-	not associated	not associated	-	-	not associated	not associated	-	-
*TNNI3K*	1p31.1	*AD*	not associated	not associated		-	limited	limited	-	-	not associated	not associated	-	-
*TRIM63*	1p36.11	*AD, AR*	not associated	not associated	-	-	not associated	not associated	-	-	moderate	limited	-	-
*VCL*	10q22.2	*AD*	not associated	not associated	-	-	moderate	moderate	-	-	not associated	limited	-	-

**Table 3 ijms-25-09787-t003:** Minor genes identified in this review, along with their clinical validity, as reported by the primary source ClinGen [45]. Pathogenic (P/LP), and VUS variants reported in ClinVar [47], and relevant animal studies found on PubMed, are provided.

Gene	Hereditary	Protein	ACM	DCM	HCM	N. P/LP Variants	N. VUS	Phenotype Submission	Animal Studies
*ABCC9*	*AD*	ATP-binding cassette subfamily C member 9	not associated	limited	not associated	9	712	DCM	not available
*ACTN2*	*AD*	Alpha-actinin 2	not associated	not associated	not associated	8	685	DCM, HCM	not available
*ANKRD1*	*AD*	Ankyrin repeat domain 1	not associated	limited	not associated	0	227	DCM, HCM	[73]
*CALR3*	*AD*	Calreticulin 3	not associated	not associated	not associated	0	145	HCM	[77,78]
*CDH2*	*AD*	Cadherin 2	limited	not associated	not associated	3	10	ACM	not available
*CSRP3*	*AD, SD*	Cysteine- and glycine-rich Protein 3	not associated	limited	moderate	16	233	DCM, HCM	[90]
*CTF1*	*AD*	Cardiotrophin 1	not associated	limited	not associated	0	3	HCM	not available
*CTNNA3*	*AD*	Catenin alpha 3	limited	not associated	not associated	3	400	ACM	[99]
*DTNA*	*AD*	Dystrobrevin alpha	not associated	limited	not associated	1	7	DCM	not available
*EYA4*	*AD*	EYA transcriptional coactivator and phosphatase 4	not associated	limited	not associated	26	445	DCM	[103]
*GATAD1*	*AR*	GATA zinc finger domain-containing 1	not associated	limited	not associated	1	75	DCM	[108]
*ILK*	*AD*	Integrin-linked kinase	not associated	limited	not associated	0	6	DCM	[112]
*JPH2*	*AD, SD*	Junctophilin 2	not associated	moderate	moderate	5	337	DCM, HCM	[120]
*KLF10*	*AD*	Krüppel-like factor 1	not associated	not associated	limited	0	1	HCM	not available
*KLHL24*	*AR*	Kelch-like family member 24	not associated	not associated	moderate	3	1	HCM	[124]
*LAMA4*	*AD*	Laminin subunit alpha 4	not associated	limited	not associated	0	704	DCM	[110]
*LDB3*	*AD*	LIM domain-binding 3	not associated	limited	not associated	3	141	DCM	[138]
*MT-TI*	*-*	Transfer RNA, mitochondrial, isoleucine	not associated	not associated	moderate	6	11	HCM	not available
*MYH6*	*AD*	Myosin heavy chain 6	not associated	limited	limited	7	983	DCM, HCM	[149]
*MYLK2*	*AD*	Myosin light chain kinase 2	not associated	not associated	not associated	0	242	HCM	not available
*MYOM1*	*AD*	Myomesin 1	not associated	not associated	not associated	0	738	HCM	not available
*MYOZ2*	*AD*	Myozenin 2	not associated	not associated	not associated	1	98	HCM	[160,162]
*MYPN*	*AD*	Myopalladin	not associated	not associated	not associated	31	646	DCM, HCM	[166]
*NEBL*	*AD*	Nebulette	not associated	limited	not associated	1	474	DCM	[274]
*NEXN*	*AD*	Nexilin F-actin-binding protein	not associated	moderate	limited	20	260	DCM, HCM	[175,178]
*NKX2-5*	*AD*	NK2 homeobox 5	not associated	limited	not associated	1	3	DCM	not available
*OBSCN*	*AD*	Obscurin	not associated	limited	limited	0	1	DCM, HCM	[193,194,195,196]
*PDLIM3*	*AD*	PDZ and LIM domain 3	not associated	not associated	limited	0	192	HCM	not available
*PLEKHM2*	*AR*	Pleckstrin homology and RUN domain-containing M2	not associated	limited	not associated	0	378	DCM	not available
*PRDM16*	*AD*	PR/SET domain 16	not associated	limited	not associated	1	30	DCM	[207,210]
*PSEN2*	*AD*	Presenilin 2	not associated	limited	not associated	0	48	DCM	[213]
*RPS6KB1*	*AD*	Ribosomal protein S6 kinase B1	not associated	not associated	limited	0	0	HCM	not available
*RYR2*	*AD*	Ryanodine receptor 2	not associated	not associated	limited	0	72	HCM	[218,223]
*SGCD*	*AD*	Sarcoglycan delta	not associated	limited	not associated	17	132	DCM	not available
*TBX20*	*AD*	T-box transcription factor 20	not associated	limited	not associated	4	9	DCM	[231,233,234]
*TCAP*	*AD*	Titin-cap	not associated	limited	not associated	20	162	DCM, HCM	not available
*TGFB3*	*AD*	Transforming growth factor beta 3	limited	not associated	not associated	4	44	ACM	not available
*TJP1*	*AD*	Tight junction protein 1	limited	not associated	not associated	0	0	ACM	not available
*TNNI3K*	*AD*	TNNI3 interacting kinase	not associated	limited	not associated	5	44	DCM	not available
*TRIM63*	*AD, AR*	Tripartite motif-containing 63	not associated	not associated	moderate	1	11	HCM	[36,259]
*VCL*	*AD*	Vinculin	not associated	moderate	not associated	3	664	DCM, HCM	[266,268,269,270,272]

## Data Availability

The authors are available to share the dataset collected for this review article.

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
