# Peer review of "Unveiling the Spectrum of Minor Genes in Cardiomyopathies: A Narrative Review"

_ijms, 2024, doi:10.3390/ijms25189787_

Round 1
Reviewer 1 Report
Comments and Suggestions for Authors
The manuscript is very thorough, very modern, very important for all cardiologists dealing with cardiomyopathies (pediatricians, adult cardiologists). Special attention is paid to arrhythmogenic cardiomyopathy of the right ventricle, dilatational cardiomyopathy and hypertrophic cardiomyopathy. The genetic analysis is presented thoroughly and comprehensively, so that - it seems that there is no knowledge in today's genetics of cardiomyopathies that is not stated in the presented publication
We would be especially pleased if the authors of this great work would include our work with the title:
Malčić I, Buljević B: Arrhythmogenic right ventricular cardiomyopathy. Naxos Island disease and Carvajal syndrome.
Central Europ Ped 2017;13(2):93-106.
Author Response
Reviewer: 1
The manuscript is very thorough, very modern, very important for all cardiologists dealing with cardiomyopathies (pediatricians, adult cardiologists). Special attention is paid to arrhythmogenic cardiomyopathy of the right ventricle, dilatational cardiomyopathy and hypertrophic cardiomyopathy. The genetic analysis is presented thoroughly and comprehensively, so that - it seems that there is no knowledge in today's genetics of cardiomyopathies that is not stated in the presented publication
We would be especially pleased if the authors of this great work would include our work with the title:
Malčić I, Buljević B: Arrhythmogenic right ventricular cardiomyopathy. Naxos Island disease and Carvajal syndrome. Central Europ Ped 2017;13(2):93-106.
REPLY: We sincerely thank this Reviewer for the positive evaluation of our manuscript. We considered the suggested paper and included it in the revised version of the manuscript (see new ref. 10, DOI:10.5457/p2005-114.177, page 2, line 75).
Reviewer 2 Report
Comments and Suggestions for Authors
The idea of the review article 'Unveiling the spectrum of minor genes in cardiomyopathies: a narrative review' submitted by Micolonghi is great. However, this manuscript misses several really important points and needs several extensions.
1.) I am wondering why restrictive cardiomyopathy was not included? Recently, the article 'Genetic insights into primary restrictive cardiomyopathy' was published. This would be a good starting point.
2.) The authors used the term ARVC instead of ACM. This is really not up to date. So many studies have shown that there are also cases with left- or biventricular arrhythmogenic cardiomyopathy. If the authors use instead of ARVC the term arrhythmogenic cardiomyopathy, there are several important manuscripts which were ignored by the authors. This should be definitely changed. The review article 'Insights into Genetics and Pathophysiology of Arrhythmogenic Cardiomyopathy' is a good starting point for this point.
3.) Please add OMIM identifiers for all genes and genetic diseases, when you introduce them the first time.
4.) Several abbreviations were explained twice. Please change.
5.) Summarizing figures would be absolutely necessary to support the text.
6.) Animal studies were frequently ignored. This should be changed, since animal models provide evidence for pathogenicity of several genes.
7.) I could not follow, why JUP and PLN were ignored. I would add short paragraphs both both genes.
8.) Line 54/55: Please add references for each gene. In addition it is necessary to add the DES gene also to ARVC/ACM. See for example 'Phenotype and Clinical outcomes in Desmin-Related Arrhythmogenic Cardiomyopathy' or 'De novo mutation N116S is associated with arrhythmogenic right ventricular cardiomyopathy'.
9.) What is with ILK and LEMD2 as ACM associated genes? For ILK, zebrafish and cell culture studies revealed definitely the impact of mutations in ACM. See 'Mutationsin ILK, encoding integrin-linked kinase, are associated with arrhythmogenic cardiomyopathy'. There are also several papers demonstrating that a missense mutation in LEMD2 could cause ACM in humans (hutterite population) and in mice. I strongly suggest to add these genes also to the list of ACM associated genes.
10.) Table 2: Please add references for each gene and update the phenotypes. For example, it is well known that mutations in DES cause also ACM and RCM and LVNC.
11.) Table 3: This table is not complete. I would add the major genes. For example DSC2, DSG2, TMEM43 and PKP2 are missing for ACM. Why? I would not ignore the major genes in summarizing tables.
12.) The impact of KLHL24 on desmin is not explained.
13.) In general the cellular and molecular details to nearly all genes are mainly ignored. Could you describe the functions in more detail?
14.) The ACMG guidelines for mutation classification should be explained. What are criteria for benign and pathogenicity? This point is completely ignored.
15.) The chromose localization of each gene should be summarized in a table or even better in a summarizing figure. This point is completely ignored.
16.) Please add a table with a mouse or rat model for each gene!
17.) Line 71/72: Please add references for each gene.
18.) Line 80/91: Please add a reference for each gene.
19.) Type of interitance should be explained in more detail.
20.) Please add references for each gene!
21.) Could you also explain the biochemical structural properties of the encoded proteins? What is the impact of mutations on the biochemical structure of the affected proteins? This point is completly ignored.
22.) Could you add a paragraph reviewing novel developments for gene therapy! For example there is a lot of progress for genes like PKP2.
In summary, the idea of this review article is great. However, this article ignores so many important papers and does not discuss important genes in detail. Therefore, I suggest a major revision for this manuscript.
Author Response
Reviewer: 2
The idea of the review article 'Unveiling the spectrum of minor genes in cardiomyopathies: a narrative review' submitted by Micolonghi is great. However, this manuscript misses several really important points and needs several extensions.
REPLY: We thank this Reviewer for the effort made on revising our manuscript and for the helpful suggestions provided to improve it. We revised the manuscript accordingly and made substantial changes to the original article. Our point-by-point replies to your comments follow.
1.) I am wondering why restrictive cardiomyopathy was not included? Recently, the article 'Genetic insights into primary restrictive cardiomyopathy' was published. This would be a good starting point.
REPLY: Thank you for pointing this out. Restrictive cardiomyopathy (RCM) is a rare cardiomyopathy that can present severe clinical manifestations and poor prognosis. We decided to not include the discussion of minor genes contributing to RCM because the genetic basis of this disease is still less understood. More specifically, the gathering sources used to prepare our article do not fully evaluate RCM, and therefore did not allow us to develop an appropriate and uniform discussion for diagnostic purposes: ClinGen, GeneReview and EMQN disease-gene curations are not yet available, and the Expert Consensus Statement cites only genes with previously definitive associations to cardiomyopathies. For the sake of completeness, we considered the suggested paper, where Brodehl and Gerull listed 19 RCM-associated genes: 14 genes are already associated with cardiomyopathies and thus were excluded from our discussion, two genes are not curated by selected sources (TMEM87B and DCBLD2), one gene (CRYAB) is associated with a syndromic condition and thus it does not meet our inclusion criteria, two genes (MYPN and ACTN2) are already included in our discussion as minor genes for DCM and HCM. For these latter genes, we added a comment to integrate their involvement also in RCM (page 17, lines 811-812 and page 9, lines 338-339, respectively).
2.) The authors used the term ARVC instead of ACM. This is really not up to date. So many studies have shown that there are also cases with left- or biventricular arrhythmogenic cardiomyopathy. If the authors use instead of ARVC the term arrhythmogenic cardiomyopathy, there are several important manuscripts which were ignored by the authors. This should be definitely changed. The review article 'Insights into Genetics and Pathophysiology of Arrhythmogenic Cardiomyopathy' is a good starting point for this point.
REPLY: Thank you for this suggestion that helps us to describe this condition more comprehensively. First, we changed the term from ARVC to ACM in the general description of the condition. Second, we used ARVC (arrhythmogenic right ventricular cardiomyopathy), ALVC (arrhythmogenic left ventricular cardiomyopathy) or BiVACM (biventricular arrhythmogenic cardiomyopathy) where required to specify the phenotypic variants of ACM. Even so, the disease-gene curation is available in the gathering sources only under the original designation of “ARVC”, which embraces also the description of the other forms. To avoid confusion for the readers, we declared this caveat (page 3, lines 144-150). Furthermore, we considered the genes mentioned by the suggested paper to confirm the appropriateness of ACM-association. Regarding both junctional and non-junctional genes, nine genes (PKP2, DSG2, DSC2, DSP, FLNC, DES, SCN5A, LMNA, TMEM43) are definitively associated with cardiomyopathies and thus were excluded, three genes (CDH2, CTNNA3, TJP1) are already included as implicated in ACM, four genes (ILK, LDB3, ACTN2, RYR2) are described in our discussion as minor genes for DCM/HCM and, in fact, we added a comment about their involvement also in ACM (page 13, lines 590-591; page 15, line 717-718; page 9, lines 338-339; page 20, lines 1015-1016, respectively). Finally, one gene is not curated by selected sources (LEMD2), and two genes (JUP and PLN) have a significant role in cardiomyopathies (although not conclusive) and thus were excluded.
3.) Please add OMIM identifiers for all genes and genetic diseases, when you introduce them the first time.
REPLY: Added, thank you.
4.) Several abbreviations were explained twice. Please change.
REPLY: Changed, thank you.
5.) Summarizing figures would be absolutely necessary to support the text.
REPLY: Thank you for this suggestion. We agree and thus we added a new Figure 1 to graphically summarize the disease-gene associations discussed in the article.
6.) Animal studies were frequently ignored. This should be changed, since animal models provide evidence for pathogenicity of several genes.
REPLY: This is a good observation. Animal studies give a great contribution to classify genes based on their disease-causative implications. In fact, animal studies were already mentioned in our previous version of the manuscript when available and relevant. To fully address your comment, we searched further the available literature, but we could not find any other study.
7.) I could not follow, why JUP and PLN were ignored. I would add short paragraphs both both genes.
REPLY: In this regard we clarify that JUP and PLN are not reported as definitive genes in all searched sources. For this reason, they were not included in our Table 1. However, they have a significant role in cardiomyopathies (as reported in both EMQN and the literature). Based on the latter consideration, they could not be qualified as minor genes and were excluded in our discussion (as stated in page 3 lines 115-121). To be consistent with our approach for minor genes definition, we did not consider appropriate adding specific paragraphs to describe JUP and PLN. We hope you may agree.
8.) Line 54/55: Please add references for each gene. In addition it is necessary to add the DES gene also to ARVC/ACM. See for example 'Phenotype and Clinical outcomes in Desmin-Related Arrhythmogenic Cardiomyopathy' or 'De novo mutation N116S is associated with arrhythmogenic right ventricular cardiomyopathy'.
REPLY: Thank you for pointing this out. DES is reported as a definitive gene for DCM in all searched sources. On this basis, it was included in Table 1 but excluded in our discussion regarding the minor genes (as declared in page 3 lines 132-134). Nevertheless, we agree that knowledge about the involvement of definitive genes in other phenotypes exists, but it is not clearly shown here. To address this issue and accomplish your request, we added a mention on known minor phenotypes for definitive genes in Table 1. However, no description was included in the text to be consistent with our main topic on rare genetic contributors to CMPs.
Moreover, we added references for each gene in the text, as suggested.
9.) What is with ILK and LEMD2 as ACM associated genes? For ILK, zebrafish and cell culture studies revealed definitely the impact of mutations in ACM. See 'Mutationsin ILK, encoding integrin-linked kinase, are associated with arrhythmogenic cardiomyopathy'. There are also several papers demonstrating that a missense mutation in LEMD2 could cause ACM in humans (hutterite population) and in mice. I strongly suggest to add these genes also to the list of ACM associated genes.
REPLY: Thank you for this comment. Regarding ILK, we added a comment about its involvement in ACM (page 13, lines 590-591). Unfortunately, LEMD2 is not considered in the questioned sources. Three genes (LDB3, ACTN2, RYR2) were integrated with additional information on their implication in ACM (see reply to point 2).
10.) Table 2: Please add references for each gene and update the phenotypes. For example, it is well known that mutations in DES cause also ACM and RCM and LVNC.
REPLY: We added references for each gene in the text, while keeping the references for the questioned sources in the tables. Moreover, we updated the phenotypes for the major genes in Table 1 (see reply to point 8). As an example, DES is now associated with ACM, whereas RCM and LVNC are not evaluated in the present article. Table 2 is already comprehensive for phenotypes of minor genes. We hope these modifications are acceptable.
11.) Table 3: This table is not complete. I would add the major genes. For example DSC2, DSG2, TMEM43 and PKP2 are missing for ACM. Why? I would not ignore the major genes in summarizing tables.
REPLY: We updated the Table 1 of the revised manuscript with info about phenotypes and variant numbers for the major genes. The summarizing Tables 2 and 3 remain fully focused on the minor genes. We hope you may be satisfied with these changes.
12.) The impact of KLHL24 on desmin is not explained.
REPLY: We updated the KLHL24 impact on desmin, as suggested, in the KLHL24 paragraph (page 14, lines 645-646).
13.) In general the cellular and molecular details to nearly all genes are mainly ignored. Could you describe the functions in more detail?
REPLY: Thank you for this comment. We believe that cellular and molecular details of all genes are very important. In fact, we have already included the molecular information at the beginning of each gene paragraph in the original version. Since the scope of our review was to provide a useful guide to clinicians for interpretation of CMPs minor genes that are nowadays screened in CMPs gene panels, we limited the description of the cellular and molecular data to few lines. In the revised manuscript we added info for those genes where details were lacking (KLHL24 and MYLK2 paragraphs).
14.) The ACMG guidelines for mutation classification should be explained. What are criteria for benign and pathogenicity? This point is completely ignored.
REPLY: Thank you for this comment. We cite the ACMG recommendations for variant classification in the revised manuscript (page 4, lines 201-207). However, explaining/assigning the criteria is well beyond the scope of this review.
15.) The chromosome localization of each gene should be summarized in a table or even better in a summarizing figure. This point is completely ignored.
REPLY: We added this information in the corresponding tables.
16.) Please add a table with a mouse or rat model for each gene!
REPLY: To accomplish your request, we added a column in Table 3 reporting the most relevant animal studies mentioned in the text.
17.) Line 71/72: Please add references for each gene.
REPLY: Added, thank you.
18.) Line 80/91: Please add a reference for each gene.
REPLY: Added, thank you.
19.) Type of interitance should be explained in more detail.
REPLY: We explained the type of inheritance (page 4, lines 238-251).
20.) Please add references for each gene!
REPLY: We added the references (see above points).
21.) Could you also explain the biochemical structural properties of the encoded proteins? What is the impact of mutations on the biochemical structure of the affected proteins? This point is completly ignored.
REPLY: We thank the reviewer for bringing up this interesting point. Our work, however, was focused on the overall importance of minor genes rather than on individual variants. Additionally, variants for most minor genes are currently classified as Variants of Uncertain Significance (VUS) according to ACMG guidelines, due to the ongoing debate and conflicting evidence regarding their functional roles.
22.) Could you add a paragraph reviewing novel developments for gene therapy! For example there is a lot of progress for genes like PKP2.
REPLY: We thank the reviewer for this interesting suggestion. We added a few lines in the revised Conclusion session, highlighting how a complete comprehension of the possible pathogenic role of these minor genes in CMPs will pave the way for therapy interventions, providing PKP2 as an example as the reviewer suggested.
In summary, the idea of this review article is great. However, this article ignores so many important papers and does not discuss important genes in detail. Therefore, I suggest a major revision for this manuscript.
REPLY: Thank you for your general comment. We appreciate your thorough review and we made the necessary revisions to address the important papers and genes that were previously overlooked, while keeping our main focus on clinical practice and diagnostic setting. Moreover, we thought it was appropriate to update our review in accordance with the newly released work by Clingen on HCM (DOI: 10.1101/2024.07.29.24311195, PMID: 39132495)
Round 2
Reviewer 2 Report
Comments and Suggestions for Authors
Congratulation. The authors have improved their manuscript. I suggest top accept it for publication.